JCB | Journal of Cell Biology

# V-ATPase/TORC1-mediated ATFS-1 translation directs mitochondrial UPR activation in *C. elegans*

Terytty Yang Li[1]*, Arwen W. Gao[1]*, Xiaoxu Li[1], Hao Li[2], Yasmine J. Liu[1], Amelia Lalou[1], Nagammal Neelagandan[3], Felix Naef[3], Kristina Schoonjans[2], and Johan Auwerx[1]

To adapt mitochondrial function to the ever-changing intra- and extracellular environment, multiple mitochondrial stress response (MSR) pathways, including the mitochondrial unfolded protein response (UPR$^{mt}$), have evolved. However, how the mitochondrial stress signal is sensed and relayed to UPR$^{mt}$ transcription factors, such as ATFS-1 in *Caenorhabditis elegans*, remains largely unknown. Here, we show that a panel of vacuolar H$^+$-ATPase (v-ATPase) subunits and the target of rapamycin complex 1 (TORC1) activity are essential for the cytosolic relay of mitochondrial stress to ATFS-1 and for the induction of the UPR$^{mt}$. Mechanistically, mitochondrial stress stimulates v-ATPase/Rheb-dependent TORC1 activation, subsequently promoting ATFS-1 translation. Increased translation of ATFS-1 upon mitochondrial stress furthermore relies on a set of ribosomal components but is independent of GCN-2/PEK-1 signaling. Finally, the v-ATPase and ribosomal subunits are required for mitochondrial surveillance and mitochondrial stress-induced longevity. These results reveal a v-ATPase-TORC1-ATFS-1 signaling pathway that links mitochondrial stress to the UPR$^{mt}$ through intimate crosstalks between multiple organelles.

## Introduction

Mitochondria are essential organelles participating in numerous cellular processes, such as energy harvesting, intermediate metabolism, calcium buffering, apoptosis, and immune response (Mishra and Chan, 2014; Nunnari and Suomalainen, 2012; West and Shadel, 2017). Although mitochondria possess their own genome, most mitochondrial proteins are encoded in the nucleus. Therefore, the expression of the mitochondrial proteome requires tight coordination between the two genomes to adapt to changes in the cellular milieu and extracellular environment (Mottis et al., 2019; Zhu et al., 2022). The mitochondrial unfolded protein response (UPR$^{mt}$), a branch of the mitochondrial stress response (MSR), is an adaptive transcriptional response that aims at resolving protein-folding stress by orchestrating the remodeling of gene expression programs after multiple forms of mitochondrial stress (Mottis et al., 2019; Shpilka and Haynes, 2018). In general, the activation of the UPR$^{mt}$ preserves fitness during aging and delays the onset of age-related-diseases (Higuchi-Sanabria et al., 2018; Lima et al., 2022; Sun et al., 2016; Vafai and Mootha, 2012).

Although first described in mammalian cells (Zhao et al., 2002), the UPR$^{mt}$ has been mainly studied in the nematode *Caenorhabditis elegans*. In *C. elegans*, the UPR$^{mt}$ is induced when the stress-activated transcription factor-1 (ATFS-1) translocates to the nucleus in response to mitochondrial perturbations (Nargund et al., 2012). In addition, a number of other transcription factors/co-factors, histone methyltransferases, demethylases, acetyltransferases, and deacetylases work together with ATFS-1 during UPR$^{mt}$ activation (Haynes et al., 2007; Li et al., 2021; Merkwirth et al., 2016; Shao et al., 2020; Tian et al., 2016; Yuan et al., 2020; Zhu et al., 2020). Despite the fact that some evidence implicated the involvement of TORC1 components in UPR$^{mt}$ signaling (Baker et al., 2012; Haynes et al., 2007; Runkel et al., 2013; Shpilka et al., 2021), these studies either did not explore in detail the molecular mechanisms involved (Baker et al., 2012; Haynes et al., 2007; Runkel et al., 2013) or primarily focused on the role of TORC1 activity in development-associated UPR$^{mt}$, which is for mitochondrial network expansion (Shpilka et al., 2021). Thus, the function of TORC1 during mitochondrial stress and the mechanism of how TORC1 mediates UPR$^{mt}$ activation remains to be revealed. In mammalian cells, mitochondrial dysfunction triggers the integrated stress response (ISR; Costa-Mattioli and Walter, 2020;

......................................................................................................................................................................................

[1]Laboratory of Integrative Systems Physiology, Interfaculty Institute of Bioengineering, École Polytechnique Fédérale de Lausanne, Lausanne, Switzerland; [2]Laboratory of Metabolic Signaling, Interfaculty Institute of Bioengineering, École Polytechnique Fédérale de Lausanne, Lausanne, Switzerland; [3]Laboratory of Computational and Systems Biology, Institute of Bioengineering, School of Life Sciences, École Polytechnique Fédérale de Lausanne, Lausanne, Switzerland.

*T.Y. Li and A.W. Gao contributed equally to this paper. Correspondence to Johan Auwerx: admin.auwerx@epfl.ch

Arwen W. Gao's present address is Laboratory Genetic Metabolic Diseases, Amsterdam Gastroenterology, Endocrinology, and Metabolism, Amsterdam UMC, University of Amsterdam, Amsterdam, The Netherlands.



Pakos-Zebrucka et al., 2016), in which the phosphorylation of the eukaryotic translation initiation factor 2α (EIF2α) results in the translation of the ATF4, ATF5, and CHOP transcription factors that jointly coordinate a gene expression program considered the functional equivalent of the UPR^mt (Mottis et al., 2019; Shpilka and Haynes, 2018). However, little is known about how the mitochondrial stress signal is transmitted through the cytosol and sensed by these UPR^mt transcription factors/co-factors. Furthermore, whether the communication between mitochondria and other cellular organelles, such as the lysosomes and ribosomes, contribute to the activation of the UPR^mt remains poorly understood.

Here, we demonstrate that stressed mitochondria increase TORC1 activity through a v-ATPase- and Rheb-dependent mechanism in *C. elegans*. Activated TORC1 thereby leads to increased translation of the UPR^mt transcription factor, ATFS-1, a process mediated by cytosolic ribosomes. The accumulated ATFS-1 protein, which is excluded from mitochondria (Nargund et al., 2012), then translocates to the nucleus and mediates the induction of a specific panel of UPR^mt effector genes. Many of these UPR^mt effectors play positive roles in the recovery of mitochondrial function, metabolic reprogramming, and lifespan extension. Collectively, our findings reveal a pivotal role of v-ATPase-TORC1-ATFS-1 signaling in UPR^mt activation and mild mitochondrial stress-induced longevity. Furthermore, the current study highlights that cytosolic relay of the mitochondrial stress signal from mitochondria to the nucleus also relies on the tight coordination of multiple cellular organelles, including lysosomes and ribosomes.

## Results

### V-ATPase mediates UPR^mt activation in *C. elegans*

We performed an RNA interference (RNAi) screen to identify genes required for UPR^mt activated by *cco-1* (*cytochrome c oxidase-1*) RNAi (Durieux et al., 2011) using the UPR^mt reporter *hsp-6p::gfp* strain of *C. elegans* (Yoneda et al., 2004). RNAi against multiple subunits of the vacuolar H^+-ATPase (v-ATPase; Lee et al., 2010), i.e., *vha-1*, *vha-4*, *vha-16*, and *vha-19*, attenuated *cco-1* (40%) RNAi-induced UPR^mt activation to a similar extent as the silencing of *atfs-1* (Nargund et al., 2012), while *vha-6*, *vha-10*, *vha-12*, and *vha-15* RNAi demonstrated a more moderate impact on UPR^mt suppression (Fig. 1, A and B). Similar effects were also found when using another mitochondrial stress inducer, i.e., the RNAi of *mrps-5* (*mitochondrial ribosomal protein S5*; Houtkooper et al., 2013; Fig. 1 C). Moreover, in agreement with previous results reported in *drp-1* mutant worms (Wei and Ruvkun, 2020), RNAi of *unc-32*, *vha-7*, *vha-14*, and *vha-18* also attenuated the UPR^mt to some extent, in response to a lower load of *cco-1* (20%) RNAi (Fig. S1 A). The suppressive effect of *vha-1* RNAi on UPR^mt was furthermore dose dependent (Fig. 1 D). Likewise, genetic or pharmacological activation of the UPR^mt by either knockdown of *spg-7*, *cts-1*, and *dlst-1*, or by treatment with antimycin A and doxycycline (Dox; Houtkooper et al., 2013; Liu et al., 2014) was abolished by *vha-1* RNAi (Fig. 1 E). Moreover, another RNAi clone (*vha-1_RNAi_2*) targeting a broader region of the *vha-1* mRNA and showing better knockdown efficiency

(Fig. S1 B) compared with the one used in the RNAi screen (*vha-1_RNAi_1*) had an even more pronounced effect in suppressing the UPR^mt (Fig. 1, F and G). As an alternative approach to inhibit v-ATPase function, we used two classical small-molecule inhibitors of v-ATPase, Bafilomycin A1 (BafA1) and Concanamycin A (ConA; Bowman et al., 1988; Drose et al., 1993). Both inhibitors abrogated UPR^mt activation induced by either *cco-1* or *mrps-5* RNAi in a concentration-dependent manner (Fig. S1, C and D). Likewise, inhibition of lysosomal acidification by using chloroquine (CQ; Homewood et al., 1972) also dose-dependently blocked the activation of the UPR^mt (Fig. 1 H). Notably, RNAi of *vha-1*, *vha-4*, *vha-16*, and *vha-19* did not affect the activation of the endoplasmic reticulum (ER) UPR (UPR^ER) induced by tunicamycin or *hsp-3* RNAi or the cytosolic UPR (UPR^CYT)/heat shock response in *C. elegans* (Fig. S1, E–G), suggesting a specific role of v-ATPase in relaying the stress signal from mitochondria.

### Transcriptional induction of UPR^mt genes requires v-ATPase subunits

Quantitative real-time PCR (qRT-PCR) revealed that *vha-1* RNAi strongly blocked the induction of multiple prototypical UPR^mt transcripts (e.g., *hsp-6*, *hsp-60*, *clec-4*, and *gpd-2*) in response to *cco-1* or *mrps-5* knockdown (Fig. 2 A), indicating that v-ATPase controls the transcriptional activation of the UPR^mt. We thus performed RNA sequencing (RNA-seq) on total RNA isolated from *hsp-6p::gfp* worms fed with *cco-1* RNAi, in combination with *vha-1*, *vha-4*, *vha-16*, *vha-19*, and *atfs-1* RNAi (Fig. 2 B and Table S1). Between 5,364 and 9,190 differentially expressed genes (DEGs; Log$_2$FC > 1 or < −1, adjusted P < 0.05) were identified in response to RNAi targeting each of the four v-ATPase subunits, and 4,563 of them were commonly regulated (Fig. S1 H). Gene ontology (GO) analysis revealed that 1,114 (24.4%) of these DEGs were related to "integral component of membrane" (Fig. S1 I), confirming a key role of v-ATPase in membrane-associated biological processes (Forgac, 2007). RNAi of *cco-1* led to the upregulation of 1,382 transcripts, among which 625 (45.2%) were dependent on at least one of the four v-ATPase subunits for induction, and 325 were dependent on all the four subunits of v-ATPase (Fig. 2 C and Table S1). These 325 genes were enriched for multiple mitochondrial pathways including "mitochondrion" (e.g., *hsp-6*), "glycolysis/gluconeogenesis" (e.g., *gpd-2*), "metabolic processes" (e.g., *idh-1*), and "transport" (e.g., *folt-1*; Fig. 2, D and E). Meanwhile, some innate immune genes, such as C-type lectin *clec-66*, were also included in this gene set (Fig. 2 D). In line with the observation that *vha-16* and *vha-19* RNAi had superior effects in suppressing the UPR^mt in *hsp-6p::gfp* worms (Fig. 1, A and C), induction of 105 UPR^mt transcripts were exclusively dependent on VHA-16 and VHA-19, but not VHA-1 and VHA-4 (Fig. 2 C). Finally, among the 625 UPR^mt transcripts regulated by the v-ATPase, 307 genes were dependent on ATFS-1 as well, covering 62.4% (307/492) of the ATFS-1-dependent program, and 318 genes were solely dependent on v-ATPase (Fig. 2 F). Interestingly, both the two gene clusters were enriched for "metabolic process," "transmembrane transport," and "carbon metabolism" (Fig. 2, G and H), confirming a vital role of v-ATPase in the rewiring of global metabolism in response to mitochondrial stress. In contrast, the transcripts of another

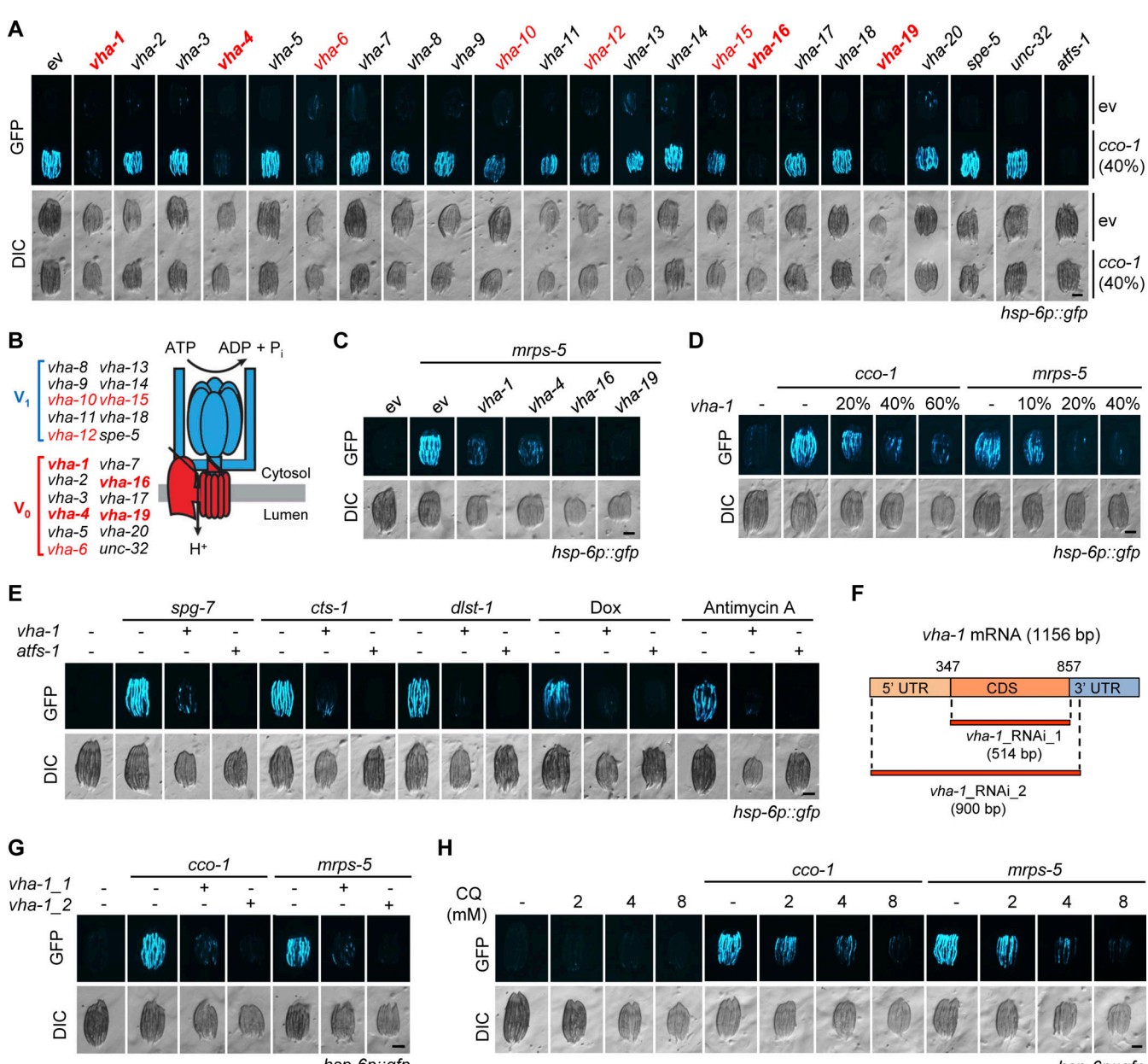

Figure 1. **Identification of a panel of v-ATPase subunits that are essential for the UPR^mt activation in *C. elegans*. (A)** RNAi of multiple v-ATPase subunits attenuated UPR^mt activation induced by *cco-1* RNAi in *hsp-6p::gfp* worms. For RNAi treatment, RNAi targeting v-ATPase subunits or *atfs-1* occupied 60%, *cco-1* RNAi occupied 40%, control RNAi (empty vector [ev]) was used to supply to a final 100% of RNAi for all conditions. DIC, differential interference contrast image. **(B)** The two functional domains of v-ATPase: the V₁ domain, which hydrolyzes ATP to generate the energy required for pumping protons, and the membrane-anchoring V₀ domain, which transports H⁺ across the lipid bilayer. Subunits whose RNAi suppressed the UPR^mt were highlighted in red, and the best four of them in red bold. **(C)** RNAi of *vha-1*, *vha-4*, *vha-16*, and *vha-19* (25%) attenuated *mrps-5* RNAi-induced UPR^mt activation in *hsp-6p::gfp* worms. **(D)** *vha-1* RNAi inhibited UPR^mt activation induced by *cco-1* or *mrps-5* (40%) RNAi in a dose-dependent manner. **(E)** *vha-1* RNAi attenuated UPR^mt activation induced by *spg-7*, *cts-1* or *dlst-1* RNAi, and by doxycycline (Dox; 30 μg/ml) or antimycin A (2.5 μM). **(F)** Schematic diagram showing the regions on mRNA targeted by the two different *vha-1* RNAi obtained from either the Vidal (*vha-1*_RNAi_1) or Ahringer library (*vha-1*_RNAi_2). **(G)** Both *vha-1* RNAi as indicated in F disrupted the UPR^mt induced by either *cco-1* or *mrps-5* RNAi. Note the more robust effect of *vha-1*_RNAi_2 in UPR^mt suppression as compared to that of *vha-1*_RNAi_1. **(H)** The lysosomal acidification inhibitor chloroquine (CQ) suppressed UPR^mt activation in a dose-dependent manner. *hsp-6p::gfp* worms were fed with control or *cco-1* (40%) RNAi, in combination with 2–8 mM CQ. Scale bars, 0.3 mm.

branch of the MSR, i.e., mitophagy/autophagy (e.g., *sqst-1* and *dct-1*), were conversely increased upon RNAi of v-ATPase subunits (Fig. S1 J). Notably, the nuclear-localized UPR^mt transcription factor ATFS-1 during mitochondrial stress (Nargund et al., 2012) was no longer detectable in the *atfs-1p::atfs-1::flag::gfp* worms fed with RNAi of *vha-1*, *vha-4*, *vha-16*, and *vha-19* (Fig. 2 I). Altogether, these results suggest that v-ATPase controls the transcriptional activation of a large set of UPR^mt genes, an effect likely achieved through the regulation of ATFS-1.

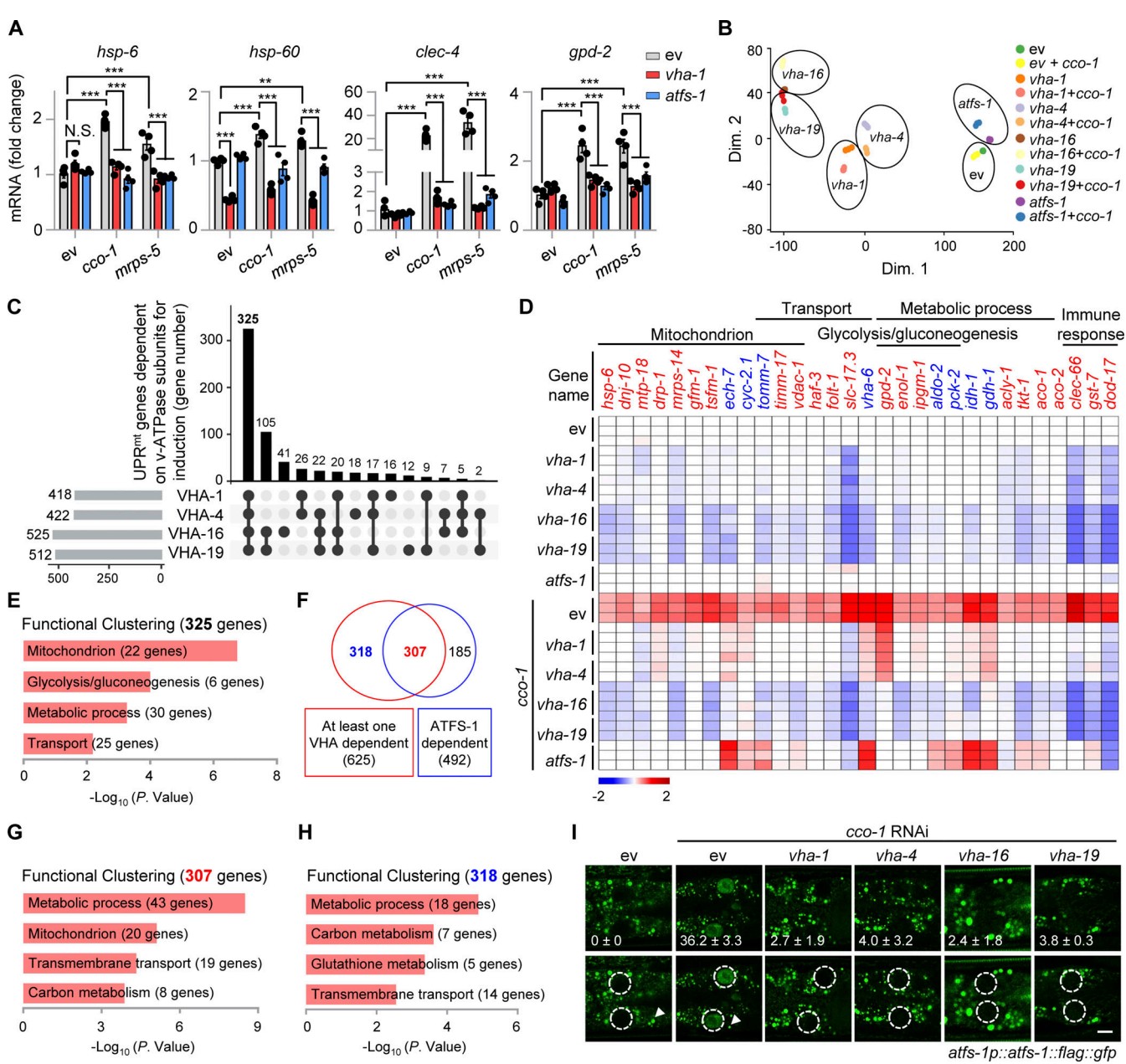

Figure 2. **Transcriptional induction of a large set of UPR^mt genes requires VHA-1, VHA-4, VHA-16, and VHA-19. (A)** qRT-PCR analysis of transcripts (*n* = 4 biologically independent samples) in *hsp-6p::gfp* worms fed with control (ev), *cco-1* or *mrps-5* (40%) RNAi in combination with *vha-1* (40%) or *atfs-1* (60%) RNAi. Error bars denote SEM. Statistical analysis was performed by ANOVA followed by Tukey post-hoc test (**P < 0.01; ***P < 0.001). **(B)** Principal-component analysis (PCA) of the RNA-seq profiles of worms fed with the indicated RNAi. **(C)** Upset plot of *cco-1* RNAi-induced UPR^mt genes that are dependent on VHA-1, VHA-4, VHA-16, or VHA-19 for induction. **(D)** Heat-maps of representative UPR^mt transcripts that are regulated by VHA-1, VHA-4, VHA-16, or VHA-19. The color-coded heat map represents gene expression differences in log₂(fold change, FC) relative to the control RNAi condition. Genes dependent on at least one VHA (−1, −4, −16 or −19) and ATFS-1 are highlighted in red. Genes dependent on at least one VHA, but not ATFS-1, are highlighted in blue. **(E)** Functional clustering of the 325 UPR^mt genes as indicated in C. **(F)** Venn diagram of the UPR^mt genes dependent on at least one VHA (−1, −4, −16 or −19), that are in common with ATFS-1-dependent UPR^mt genes, based on the RNA-seq dataset. **(G and H)** Functional clustering of the 307 (G) and 318 (H) UPR^mt genes as indicated in F. **(I)** Photomicrographs of the most proximal two intestinal cells in *atfs-1p::atfs-1::flag::gfp* worms fed with control or *cco-1* (40%) RNAi, in combination with *vha-1, vha-4, vha-16,* or *vha-19* (25%) RNAi. The nuclei were outlined with white dashed-line circles (bottom panels). The punctae (white arrowheads) are endogenous autofluorescence lysosomes. Mean percentages (±SEM) of worms with nuclear accumulation of ATFS-1::GFP are indicated (*n* = 3 independent experiments). Scale bar, 10 μm.

## Mitochondrial stress induces TORC1 activation and increases ATFS-1 expression

To further explore the molecular mechanism of how v-ATPase regulates the UPR^mt, we examined the expression of ATFS-1 (Nargund et al., 2015; Nargund et al., 2012) in *atfs-1p::atfs-1::flag::gfp* worms fed with *vha-1, vha-4, vha-16,* and *vha-19* RNAi in the absence or presence of mitochondrial stress. In worms fed with *cco-1* RNAi, the protein level of ATFS-1 was increased by more

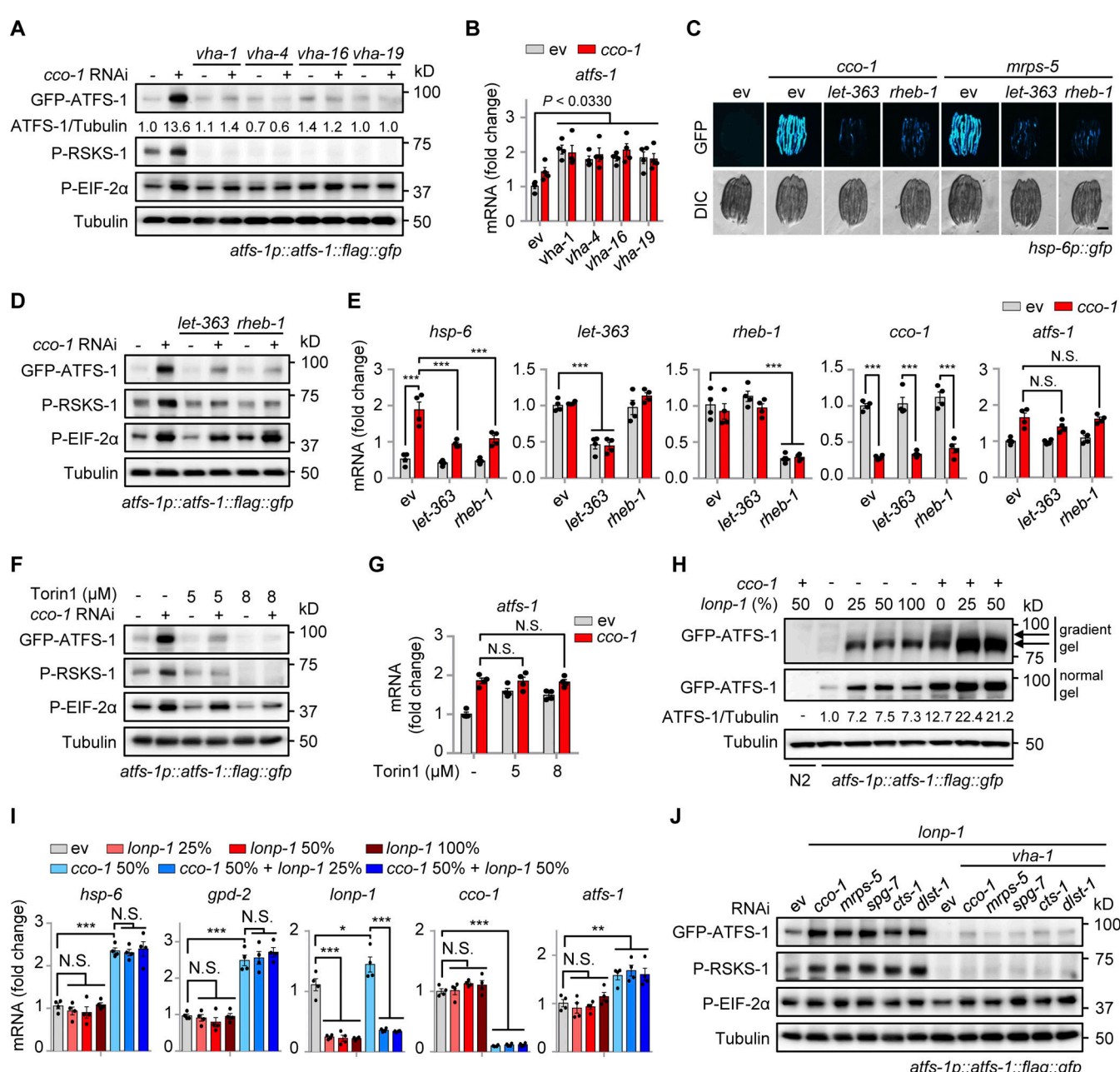

Figure 3. **Mitochondrial stress induces v-ATPase-dependent TORC1 activation and increases ATFS-1 expression. (A and B)** Western blots (A) and *atfs-1* mRNA levels (*n* = 4 biologically independent samples; B) of *atfs-1p::atfs-1::flag::gfp* worms fed with control, *vha-1, vha-4, vha-16,* or *vha-19* (25%) RNAi with or without *cco-1* (50%) RNAi. **(C)** RNAi of *let-363* or *rheb-1* attenuated UPR[mt] activation. *hsp-6p::gfp* worms were fed with control, *let-363* (75%), or *rheb-1* (75%) RNAi, in combination with *cco-1* (25%) or *mrps-5* (25%) RNAi. Scale bar, 0.3 mm. **(D and E)** Western blots (D) and qRT-PCR analysis of transcripts (*n* = 4 biologically independent samples; E) of *atfs-1p::atfs-1::flag::gfp* worms fed with control, *let-363* (60%) or *rheb-1* (60%) RNAi, with or without *cco-1* (40%) RNAi. **(F and G)** Western blots (F) and qRT-PCR analysis of transcripts (*n* = 4 biologically independent samples; G) of *atfs-1p::atfs-1::flag::gfp* worms fed with control or *cco-1* (40%) RNAi, co-treated with or without 5 or 8 μM Torin1. **(H and I)** Western blots (H) and qRT-PCR analysis of transcripts (*n* = 4 biologically independent samples; I) in *atfs-1p::atfs-1::flag::gfp* worms fed with control, *lonp-1* and/or *cco-1* RNAi. Wild-type (N2) worms fed with *cco-1* RNAi were used as a negative control. Western blots were performed with either normal or 4–12% gradient gels. **(J)** Western blots of *atfs-1p::atfs-1::flag::gfp* worms fed with control, *cco-1, mrps-5, spg-7,* or *cts-1* (50%) RNAi in the presence of *lonp-1* (25%) RNAi, and/or *vha-1* (25%) RNAi. Error bars denote SEM. Statistical analysis was performed by ANOVA followed by Tukey post-hoc test (*P < 0.05; **P < 0.01; ***P < 0.001). Source data are available for this figure: SourceData F3.

than tenfold compared with the unstressed condition, and this response was almost completely blocked by *vha-1, vha-4, vha-16,* and *vha-19* RNAi (Fig. 3 A). The v-ATPase has been shown to not only act as an ATP-driven proton pump to maintain the acidic environment of the lysosomal lumen but also as an essential

mediator for the activation of pathways, including mTORC1 signaling (Lawrence and Zoncu, 2019; Shimobayashi and Hall, 2014; Wolfson and Sabatini, 2017; Zoncu et al., 2011). We found that phosphorylation of RSKS-1, the *C. elegans* ortholog of mammalian S6 kinase (S6K) and a common readout of mTORC1

activity, was robustly increased in a v-ATPase-dependent manner in response to mitochondrial stress (Fig. 3 A). By contrast, mitochondrial stress-induced phosphorylation of EIF-2α (Baker et al., 2012), the worm ortholog of mammalian EIF2α, was only modestly decreased (upon *vha-1*, *vha-4*, and *vha-19* RNAi) or unaffected (upon *vha-16* RNAi; Fig. 3 A). Meanwhile, the *atfs-1* mRNA level was even higher in v-ATPase RNAi-fed worms compared with that in control worms (Fig. 3 B). Unlike ATFS-1, the protein level of DVE-1 (Haynes et al., 2007), another key transcription factor of the UPR^mt, was not altered by mitochondrial stress or *vha-1* RNAi (Fig. S2 A). Additionally, *vha-1* silencing conversely lead to more protein expression of XBP-1 (Calfon et al., 2002; Fig. S2 B), an essential transcription factor of the UPR^ER.

Similar to the effects of v-ATPase RNAi, suppression of worm TORC1 activity by either RNAi of *let-363* (the worm ortholog of human *mTOR*) and *rheb-1* (the worm ortholog of mTORC1 upstream activator, *Rheb, Ras homolog enriched in brain*; Inoki et al., 2003), or by applying the TORC1 catalytic inhibitor Torin1 (Thoreen et al., 2009) attenuated UPR^mt induced by *cco-1* or *mrps-5* knockdown (Fig. 3 C and Fig. S2 C). Notably, we found that ATFS-1 accumulation and RSKS-1 phosphorylation in response to *cco-1* RNAi were abrogated by RNAi of *let-363* or *rheb-1* and by Torin1 treatment, while EIF-2α phosphorylation and *atfs-1* mRNA expression were barely affected (Fig. 3, D–G). Consistent with the importance of lysosomes in TORC1 activation (Lawrence and Zoncu, 2019), the lysosomal acidification inhibitor CQ attenuated the induction of UPR^mt genes, TORC1 activity (Fedele and Proud, 2020; Jewell et al., 2015), and the accumulation of ATFS-1 in a dose-dependent manner (Fig. S2, D and E), confirming that the intact lysosomal function/pH is essential for TORC1 and UPR^mt activation. Moreover, *let-363* or *rheb-1* RNAi did not affect the activation of the UPR^ER or UPR^CYT (Fig. S2, F and G). Intriguingly, knockout of *raga-1*, the sole worm ortholog of the evolutionarily conserved *ras-related GTPase RagA* and *RagB* that are essential for the activation of the mTORC1 by exogenous amino acids (Sancak et al., 2008; Schreiber et al., 2010), lead to even more robust induction of UPR^mt genes in response to *cco-1* RNAi (Fig. S2 H), suggesting that mitochondrial stress probably represents a unique intrinsic signal for TORC1 activation independent of the Rag GTPase (Hesketh et al., 2020).

The protein level of ATFS-1 has been reported to be mainly controlled by the mitochondrial Lon protease homolog (LONP-1; Nargund et al., 2012; Shpilka et al., 2021; Yang et al., 2022), which mediates the degradation of ATFS-1 in the mitochondrial matrix so that little ATFS-1 is available for the transcriptional activation of the UPR^mt in unstressed worms. Indeed, *lonp-1* RNAi led to the upregulation of ATFS-1 protein without affecting its mRNA level and UPR^mt activity (Fig. 3, H and I). However, more accumulation of ATFS-1 protein in response to *cco-1* RNAi was detected even in the background of 50% of *lonp-1* RNAi (Fig. 3 H), a condition when *lonp-1* transcript was firmly knocked down (Fig. 3 I). As expected, *lonp-1* RNAi only increased the accumulation of mitochondrial-localized ATFS-1 (the lower band, produced after the cleavage of its mitochondrial targeting sequence; Nargund et al., 2012), while *cco-1* RNAi led to the accumulation of both the mitochondrial-localized and unprocessed

forms of ATFS-1 (Fig. 3 H). Importantly, RNAi of *vha-1*, *vha-4*, *vha-16*, and *vha-19* blocked the accumulation of ATFS-1 induced by mitochondrial stress when *lonp-1* was silenced (Fig. S2 I). Likewise, other mitochondrial stress inducers, such as *mrps-5*, *spg-7*, *cts-1*, and *dlst-1* RNAi, also increased ATFS-1 expression and RSKS-1 phosphorylation in the background of *lonp-1* RNAi, and these responses were furthermore abolished by *vha-1* RNAi (Fig. 3 J). These results suggest that v-ATPase regulates ATFS-1 protein expression upstream of LONP-1 and through a previously uncharacterized LONP-1-independent mechanism.

## Knockdown of ribosomal subunits blocks the UPR^mt and ATFS-1 translation

To test whether increased accumulation of ATFS-1 protein upon mitochondrial stress is caused either by the direct increase of *atfs-1* translation or by the suppression of other yet unknown ATFS-1 degradation pathways (e.g., through a ULP-4-mediated SUMOylation mechanism [Gao et al., 2019]), we first knocked down a set of ribosomal subunits, including the small (*rps-8* and *rps-10*) and large ones (*rpl-14*, *rpl-25.1*, *rpl-27*, *rpl-36* and *rpl-43*) in the UPR^mt reporter *hsp-6p::gfp* strain. RNAi of each individual ribosomal subunit remarkably blocked *cco-1* or *mrps-5* RNAi-induced UPR^mt activation, while tunicamycin-induced UPR^ER was only partially attenuated, and the UPR^CYT/heat shock response was not affected (Fig. 4 A and Fig. S3, A–C). In support of these results, paraquat-induced UPR^mt was also reported to rely on multiple ribosomal subunits in a previous screen study (Runkel et al., 2013). Importantly, RNAi of *rps-8*, *rps-10*, *rpl-27*, and *rpl-36* almost completely blocked ATFS-1 expression in the *atfs-1p::atfs-1::flag::gfp* strain upon mitochondrial stress (Fig. 4 B), while, in apparent contrast, the mRNA level of *atfs-1* was even upregulated in response to their knockdown (Fig. 4 C). Next, we took advantage of the *atfs-1* reporter strain *atfs-1p::H1-wCherry* (Murray et al., 2012), which was constructed such that the expression of the Histone–mCherry reporter protein is under the strict control of the upstream intergenic sequences (including both the promoter and 5′-UTR regions) of *atfs-1*. Thus, the expression/translation of this Histone–mCherry is controlled in a similar fashion as that of the endogenous ATFS-1 protein. In addition, since the translated Histone–mCherry is not degraded by LONP-1 or other ATFS-1-specific-targeting enzymes, the *atfs-1p::Histone-wCherry* transgenic strain is therefore an ideal system to study ATFS-1 translation regulation, independent of its degradation. Similar to the results acquired with the *atfs-1p::atfs-1::flag::gfp* worms (Fig. 3, A and D; and Fig. 4 B), we found that *cco-1* RNAi led to a robust upregulation of the Histone–mCherry reporter protein in the *atfs-1p::H1-wCherry* worms, which was almost completely blocked by RNAi targeting *let-363*, *rheb-1*, v-ATPase, or the ribosomal subunits (Fig. 4, D–F). Finally, we applied polysome profiling whereby free ribosomal subunits, monosomes (mRNA with one ribosome associated), and polysomes (mRNA with two or more ribosomes associated, the highly translated ribosome-mRNA fraction), were separated over a sucrose density gradient and quantified by optical density. Similar to the results as shown for *mrps-5* RNAi treatment (Molenaars et al., 2020), *cco-1* RNAi led to a shift from polysomes to monosomes (Fig. 4 G), confirming an adaptive

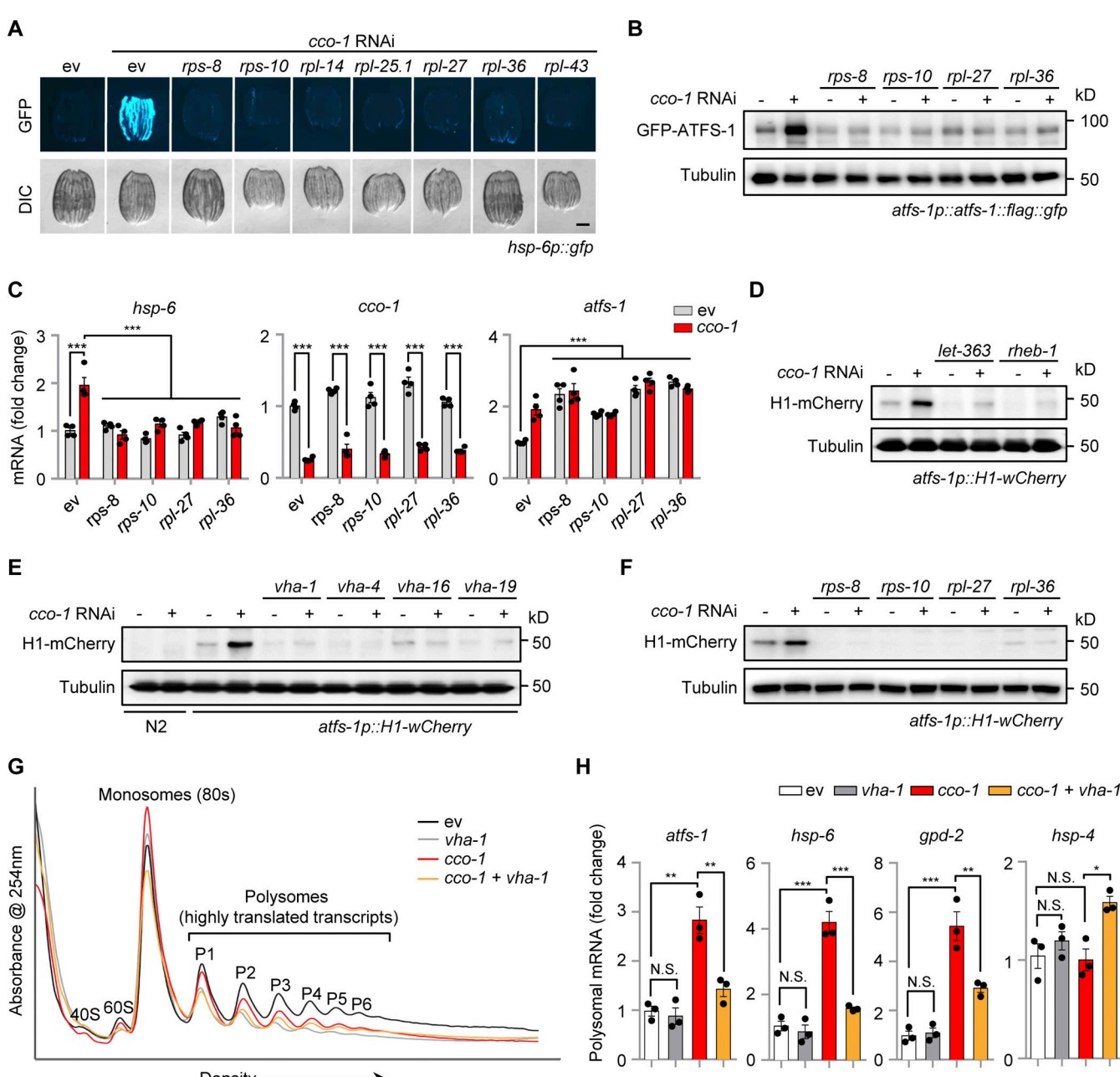

**Figure 4.** **Knockdown of ribosomal subunits blocks UPR^mt activation and ATFS-1 translation upon mitochondrial stress. (A)** RNAi of multiple ribosomal subunits attenuated UPR^mt activation induced by *cco-1* RNAi. RNAi targeting *cco-1* occupies 40%, RNAi targeting ribosomal subunits occupies 60%. Scale bar, 0.3 mm. **(B and C)** Western blots (B) and qRT-PCR analysis of transcripts (n = 4 biologically independent samples; C) in *atfs-1p::atfs-1::flag::gfp* worms fed with control, *rps-8*, *rps-10*, *rpl-27* or *rpl-36* (60%) RNAi with or without *cco-1* (40%) RNAi. **(D–F)** Western blots of *atfs-1p::H1-wCherry* worms fed with control or *cco-1* (40%) RNAi, in combination with *let-363* or *rheb-1* (60%) RNAi (D), *vha-1, vha-4, vha-16,* or *vha-19* (25%) RNAi (E), or *rps-8, rps-10, rpl-27* or *rpl-36* (60%) RNAi (F). **(G)** Representative polysome profiles of worms fed with control or *cco-1* (50%) RNAi, with or without *vha-1* (25%) RNAi. The relative positions of the ribosome subunits (40 and 60 S), monosomes (80 S), and polysomes (P1-P6) are indicated. **(H)** qRT-PCR analysis of transcripts (n = 3 biologically independent samples) in the polysomal fractions (highly translated) of worms as indicated in G. Error bars denote SEM. Statistical analysis was performed by ANOVA followed by Tukey post-hoc test (*P < 0.05; **P < 0.01; ***P < 0.001). Source data are available for this figure: SourceData F4.

cytosolic translation reduction in response to mitochondrial stress (D'Amico et al., 2017; Molenaars et al., 2020; Suhm et al., 2018). Importantly, increased polysomal mRNA of *atfs-1*, as well as that of UPR^mt targets *hsp-6* and *gpd-2*, was detected in response to *cco-1* RNAi, a phenomenon which is strongly attenuated with *vha-1* RNAi co-treatment (Fig. 4 H). Of note, the overall cellular mRNA level of *atfs-1* was rather higher on *vha-1* RNAi (Fig. 3 E). In contrast, the polysomal mRNA of other transcription factors or typical reporter genes (e.g., *hsp-4, dve-1, xbp-1s, hsf-1,* and *daf-16*) involved in different stress responses was not affected by mitochondrial stress and was either unchanged or even upregulated after the co-treatment of *vha-*

*1* RNAi (Fig. 4 H and Fig. S3 D). Of note, neither *vha-1* RNAi nor mitochondrial stress altered the overall total and polysomal mRNA levels of *rsks-1* (Fig. S3, E and F). Together, these results suggest that increased translation of *atfs-1*, mediated by v-AT-Pase/TORC1 and cytosolic ribosomes, is a key mechanism that leads to the accumulation of ATFS-1 protein for UPR<sup>mt</sup> activation in response to mitochondrial stress.

### Mitochondrial stress-induced ATFS-1 is independent of the GCN-2/PEK-1 signaling

In mammals, phosphorylation of EIF2α by four dedicated kinases (GCN2, PERK, HRI, and PKR) serves to attenuate the general cytosolic translation in response to a variety of intra- and extracellular stresses (e.g., amino-acid starvation, viral infection, oxidative and unfolded protein stress), and meanwhile stimulates the translation of ATF4, ATF5, and CHOP, the mammalian functional orthologs of ATFS-1, to activate the ISR (Costa-Mattioli and Walter, 2020; Pakos-Zebrucka et al., 2016; Quiros et al., 2017). We thus questioned whether a similar mechanism also exists in *C. elegans*. Surprisingly, RNAi of *gcn-2* and/or *pek-1*, which encode the only two known corresponding worm EIF-2α kinases, GCN-2 and PEK-1 (Baker et al., 2012; Shen et al., 2001), failed to block *cco-1* RNAi-induced UPR<sup>mt</sup> in *hsp-6p::gfp* worms (Fig. 5 A). Meanwhile, EIF-2α phosphorylation was attenuated by either *gcn-2* or *pek-1* RNAi, in both basal and mitochondrial stress conditions (Fig. 5 B). Moreover, although only GCN-2 is required for EIF-2α phosphorylation in the *clk-1(qm30)* mitochondrial mutant (Baker et al., 2012; Lakowski and Hekimi, 1996), less of EIF-2α phosphorylation was detected with the cosilencing of both *gcn-2* and *pek-1* upon *cco-1* RNAi (Fig. 5 B). In line with the results acquired in *hsp-6p::gfp* worms (Fig. 5 A), mitochondrial stress-induced upregulation of ATFS-1 was not affected by either *gcn-2*, *pek-1*, or *eif-2α* RNAi (Fig. 5, B and C). Furthermore, comparable levels of RSKS-1 phosphorylation and UPR<sup>mt</sup> transcript induction upon *cco-1* RNAi were found in the *gcn-2* or *pek-1* knockout worm mutants, even in conditions with full suppression of EIF-2α phosphorylation, as compared with that in WT (N2) worms (Fig. 5, D–F). Likewise, activation of the UPR<sup>mt</sup> was also not affected in autophagy-defective mutants (Fig. S4). Collectively, these results suggest that increased expression of ATFS-1 in response to mitochondrial stress is independent of the GCN-2/PEK-1 signaling as well as the autophagic process per se.

### A crucial role of v-ATPase and ribosomal subunits in mitochondrial surveillance

To explore the functions of v-ATPase and ribosomal subunits in mitochondrial homeostasis and adaptations upon stresses, we raised WT and the mitochondrial respiration mutants *isp-1(qm150)* and *clk-1(qm30)* (Feng et al., 2001; Lakowski and Hekimi, 1996), and on control, *vha-1*, *vha-4*, *vha-16*, *vha-19*, *rps-8*, *rps-10*, *rpl-27*, or *rpl-36* RNAi bacteria. Compared to *C. elegans* fed with control RNAi, RNAi targeting v-ATPase or ribosomal subunits led to severe synthetic growth defects of the mitochondrial stressed mutants, whereas the development of WT worms was only slightly delayed (Fig. 6 A and Fig. S5 A). Similar results were also found in worms fed with *vha-1* RNAi and/or *cco-1* RNAi

(Fig. S5 B). Thus, mitochondrial respiration mutants heavily rely on v-ATPase and ribosomal subunits to maintain growth. We then questioned whether v-ATPase and ribosomal subunits also contribute to mild mitochondrial stress-induced lifespan extension in *C. elegans* (Durieux et al., 2011; Houtkoeper et al., 2013). In line with an essential role of the v-ATPase subunits in the UPR<sup>mt</sup> (Fig. 1, A–C), RNAi of *vha-1*, *vha-4*, *vha-16*, and *vha-19* strongly attenuated the lifespan extension induced by *cco-1* or *mrps-5* RNAi (Fig. 6, B and C). Consistently, the silencing of ribosomal subunits including *rps-8*, *rps-10*, *rpl-27*, and *rpl-36* also blunted the lifespan extension induced by *cco-1* RNAi (Fig. 6 D). Thus, v-ATPase and ribosomal components play a crucial role in mitochondrial surveillance and regulate the longevity induced by mild mitochondrial stress in *C. elegans*.

## Discussion

The UPR<sup>mt</sup> was initially defined as a transcriptional response triggered by the presence or accumulation of unfolded proteins/peptides from mitochondria (Shpilka and Haynes, 2018; Zhao et al., 2002). In *C. elegans*, almost all the well-characterized UPR<sup>mt</sup> regulators, such as ATFS-1 (Nargund et al., 2012), DVE-1 (Haynes et al., 2007), MET-2/LIN-65 (Tian et al., 2016), JMJD-3.1/JMJD-1.2 (Merkwirth et al., 2016), SET-6/BAZ-2 (Yuan et al., 2020), HDA-1 (Shao et al., 2020), NuRD (Zhu et al., 2020), and CBP-1 (Li et al., 2021), are localized or translocated in the nucleus upon mitochondrial stress. However, how the mitochondrial stress signal is sensed in the cytosol and relayed to these UPR<sup>mt</sup> regulators is only partially elucidated. Moreover, whether other protein complexes and organelles, such as the lysosomes and ribosomes, also play a role in the UPR<sup>mt</sup> activation process remains elusive.

Here, we demonstrated that the mitochondrial stress is transduced through a v-ATPase-TORC1-ATFS-1 signaling pathway in the cytosol involving distinct organelles (Fig. 7). In this signaling network, stressed mitochondria increase TORC1 activity through a v-ATPase- and Rheb-dependent mechanism. Activated TORC1 thereby leads to increased translation of the UPR<sup>mt</sup> transcription factor, ATFS-1, which is dependent on the cytosolic ribosomes (Fig. 7; indicated by mechanism 1). The accumulated ATFS-1 protein then translocates to the nucleus and mediates the induction of a specific panel of UPR<sup>mt</sup> effector genes. Many of these UPR<sup>mt</sup> effectors play positive roles in the recovery of mitochondrial function, metabolic reprogramming, and lifespan extension. Importantly, genetic or pharmacological disruption of any components in this pathway robustly suppressed the UPR<sup>mt</sup>, but not other similar stress responses, such as UPR<sup>ER</sup> and UPR<sup>CYT</sup>. Our work thus reveals that in addition to the attenuated mitochondrial import of ATFS-1 in response to mitochondrial stress (Nargund et al., 2012; Fig. 7; indicated by mechanism 2), v-ATPase/TORC1-mediated upregulation of ATFS-1 translation is also essential to ensure that enough ATFS-1 is translocated to the nucleus for UPR<sup>mt</sup> activation. Of note, v-ATPase/TORC1-mediated translation of ATFS-1 seems to be a prerequisite step for the increased accumulation and the subsequent nuclear localization of ATFS-1 for UPR<sup>mt</sup> activation during mitochondrial stress, as evidenced by the GFP-ATFS-1

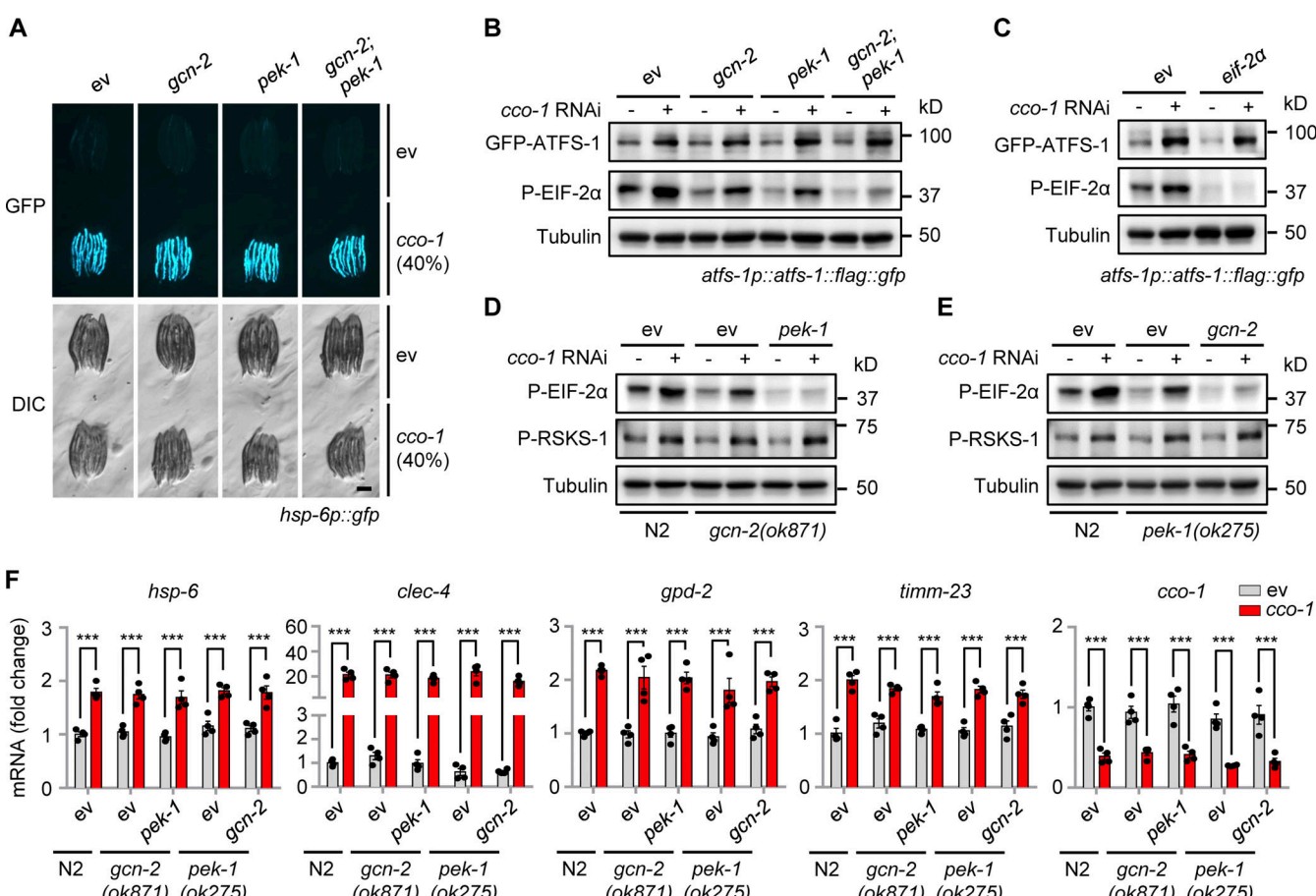

Figure 5. **Increased expression of ATFS-1 in response to mitochondrial stress is independent of the GCN-2/PEK-1 signaling. (A)** RNAi of *gcn-2* and/or *pek-1* did not affect UPR^mt activation. *hsp-6p::gfp* worms were fed with control, *gcn-2*, and/or *pek-1* RNAi, in combination with *cco-1* RNAi. Scale bar, 0.3 mm. **(B)** Western blots of *atfs-1p::atfs-1::flag::gfp* worms fed with control, *gcn-2*, and/or *pek-1* RNAi, with or without *cco-1* RNAi. **(C)** Western blots of *atfs-1p::atfs-1::flag::gfp* worms fed with control, *eif-2α* and/or *cco-1* RNAi. **(D)** Western blots of WT (N2) or *gcn-2(ok871)* worms fed with control, *pek-1,* and/or *cco-1* RNAi. **(E)** Western blots of wild-type or *pek-1(ok275)* worms fed with control, *gcn-2*, and/or *cco-1* RNAi. **(F)** qRT-PCR analysis of transcripts (*n* = 4 biologically independent samples) in wild-type, *gcn-2(ok871)* or *pek-1(ok275)* worms fed with control, *pek-1, gcn-2* and/or *cco-1* RNAi. Error bars denote SEM. Statistical analysis was performed by ANOVA followed by Tukey post-hoc test (***P < 0.001). Source data are available for this figure: SourceData F5.

nuclear localization results (Fig. 2 I), supporting that the two mechanisms likely act as a whole in MSR.

Consistent with a central role of TORC1 signaling in the UPR^mt, TORC1 and RSKS-1 were reported to be indispensable for the increased UPR^mt activity to support mitochondrial network expansion during development, a condition when TORC1 activity is already known to be active (Shpilka et al., 2021). How the TORC1 activity is activated in response to mitochondrial stress remains an important direction for future work. One possibility is that the unfolded proteins/peptides produced upon mitochondrial stress could somehow be transported from the mitochondria to the lysosomes and thereby digested to amino acids within the lysosomes, which could then lead to TORC1 activation at the lysosomal surface (Lawrence and Zoncu, 2019; Shimobayashi and Hall, 2014; Wolfson and Sabatini, 2017; Zoncu et al., 2011). We found that the *cco-1* RNAi-induced UPR^mt activation is independent of *raga-1* (Fig. S2 H), suggesting that mitochondrial stress likely represents a unique intrinsic signal for TORC1 activation by lysosome-derived amino acids (Hesketh et al., 2020), which apparently differs from what is observed

during development or upon stimulation with exogenous amino acids (Sancak et al., 2008; Schreiber et al., 2010). In support of this model, a Rab5-mediated mitochondrion-endosome-lysosome pathway functions in mitochondrial quality control and is activated upon mitochondrial dysfunction, independent of the autophagic process (Hammerling et al., 2017; Sugiura et al., 2014). More mitochondrial proteins/peptides were also detected in Rab5-positive endosomes in response to mitochondrial stress (Hammerling et al., 2020). Of note, stressed mitochondria might also directly communicate with lysosomes via mitochondria–lysosome membrane contact sites (Wong et al., 2019). Finally, v-ATPase has been also found to participate in endosomal membrane fusion processes (Peters et al., 2001), and it may thereby facilitate the transportation of mitochondria-derived unfolded proteins/peptides to the lysosomes, together with Rab5 and other cofactors.

In addition to the vital role of v-ATPase/mTOR signaling upon mitochondrial stress, we have noticed that the basal expression of some UPR^mt transcripts (e.g., *hsp-60*) decreased after the RNAi of certain v-ATPase subunits (Fig. 2, A and D; and

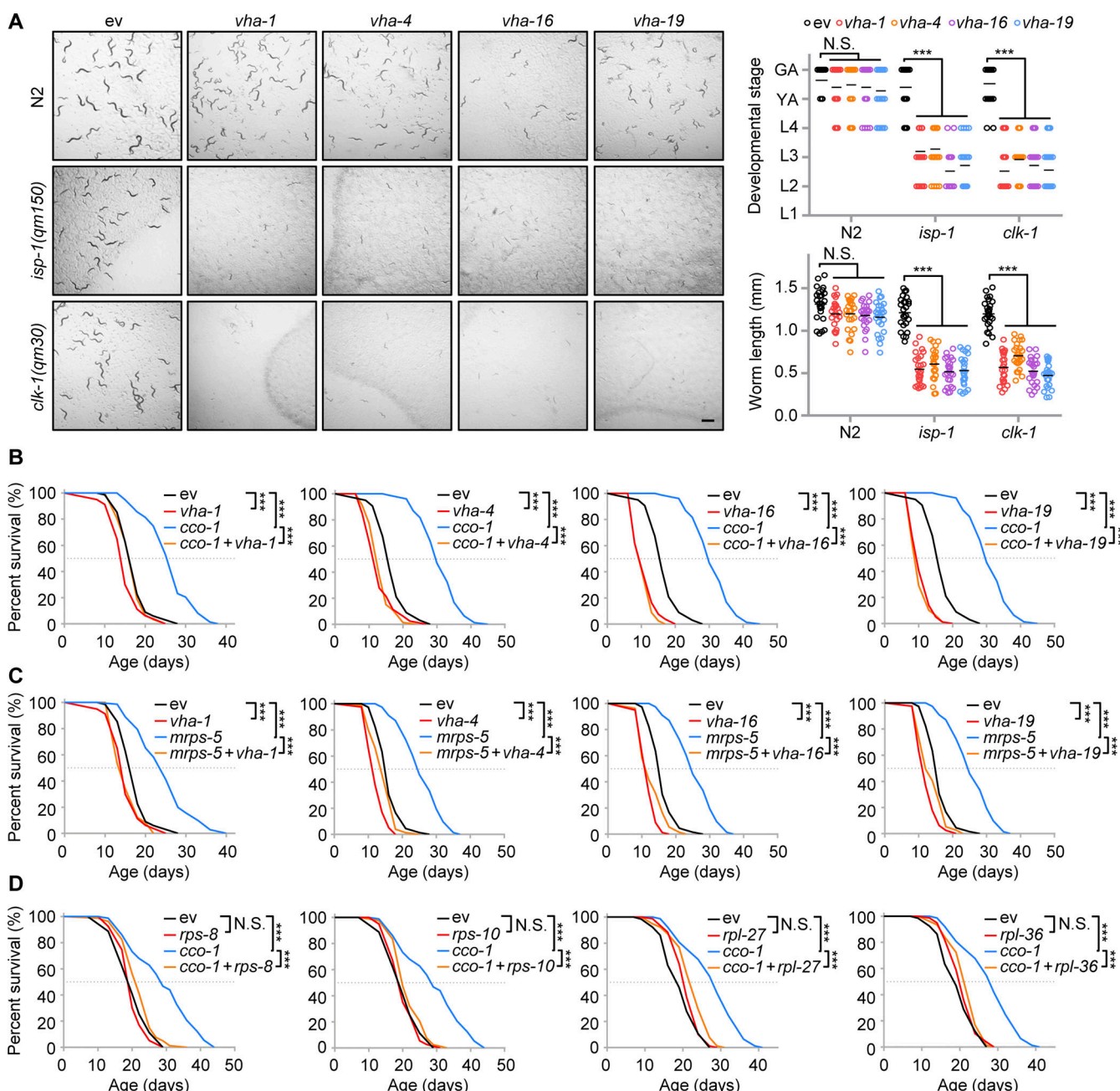

Figure 6. **A crucial role of v-ATPase and ribosomal subunits in mitochondrial surveillance and mitochondrial stress-induced longevity. (A)** Representative brightfield pictures of wild-type (N2), *isp-1(qm150),* or *clk-1(qm30)* worms fed with control (ev), *vha-1* (20%), *vha-4* (20%), *vha-16* (10%), or *vha-19* (20%) RNAi since maternal L4 stage. The developmental stage and body length of the F1 progeny were quantified on Day 4 after hatching (n = 25 worms for each condition). Scale bar, 1 mm. GA, gravid adult; YA, young adult; L1-4, larval stage 1–4. **(B and C)** RNAi of v-ATPase subunits attenuated mitochondrial stress-induced lifespan extension. Survival of worms fed with control, *vha-1* (20%), *vha-4* (20%), *vha-16* (10%), or *vha-19* (20%) RNAi, with or without *cco-1* (50%; B) or *mrps-5* (50%; C) RNAi. **(D)** RNAi of ribosomal subunits attenuates mitochondrial stress-induced lifespan extension. Survival of worms fed with control, *rps-8* (60%), *rps-10* (60%), *rpl-27* (60%), or *rpl-36* (60%) RNAi, with or without *cco-1* (40%) RNAi. Statistical analysis was performed by log-rank test (**P < 0.01; ***P < 0.001).

Table S1), indicating that v-ATPase functions in maintaining basal UPR^mt activity as well. Nevertheless, the difference between basal and stress conditions is somehow artificial, especially considering that cells and organisms are constantly exposed to various intra- and extracellular cues, and different wild *C. elegans* strains differ at the level of UPR^mt activity under basal conditions (Yin et al., 2017).

Key components in the v-ATPase-TORC1-ATFS-1 pathway are well conserved in mammals, suggesting that a similar mechanism may also exist in mammalian cells. As a case in point, mTORC1 activity has been shown to be essential for ATF4 activation downstream of growth signals (Ben-Sahra et al., 2016; Torrence et al., 2021). Further studies are therefore required to explore whether genetically or pharmacologically targeting this

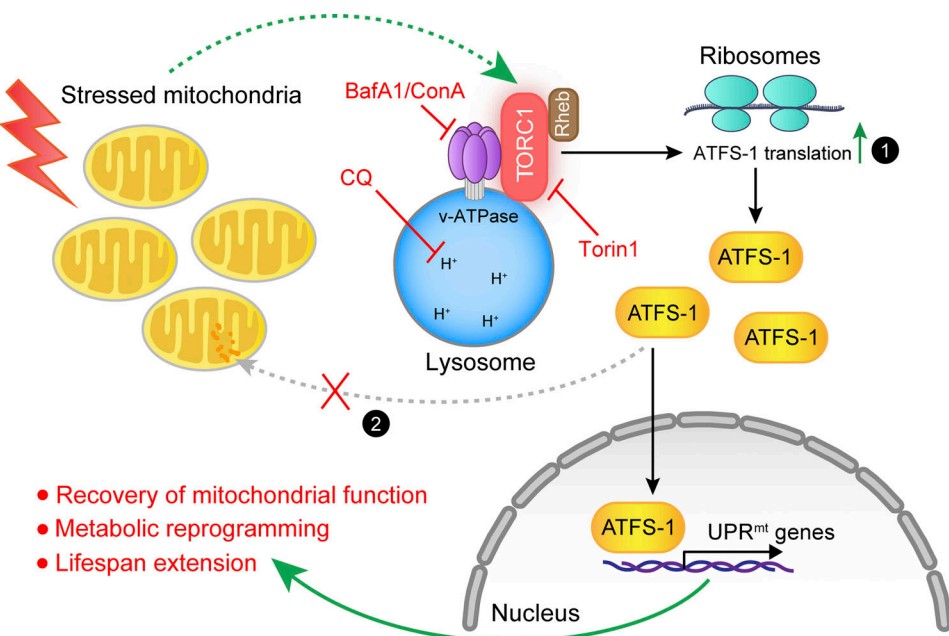

**Figure 7. Model for v-ATPase/TORC1-mediated regulation of ATFS-1 translation and UPR^mt activation in *C. elegans*.** When mitochondria are stressed in response to various intracellular or extracellular stimuli, the activity of TORC1 is increased through a v-ATPase- and Rheb-dependent mechanism, which could be blocked by both inhibitors of v-ATPase, Bafilomycin A1 (BafA1), and Concanamycin A (ConA); the lysosomal acidification inhibitor, chloroquine (CQ); as well as the TORC1 inhibitor, Torin1. Activated TORC1 thereby stimulates cytosolic ribosomes to translate the UPR^mt transcription factor ATFS-1 (mechanism 1). Under the basal nonstressed condition, ATFS-1 is transported and degraded in the mitochondria; during mitochondrial stress, in addition to increased ATFS-1 translation, mitochondrial import efficiency is also decreased (mechanism 2; Nargund et al., 2012). Both mechanisms together result in nuclear accumulation of ATFS-1, where it induces a diverse panel of UPR^mt effector genes. Many of these UPR^mt effectors play positive roles in mitochondrial function recovery, metabolic reprogramming, and lifespan extension.

v-ATPase/TORC1-driven MSR pathway could have therapeutic applications against mitochondria-associated diseases, metabolic disorders, as well as normal aging in other organisms.

**Limitations of the study**
While our current study reveals an indispensable role of v-AT-Pase/TORC1-mediated ATFS-1 translation in UPR^mt activation and mitochondrial stress-associated lifespan extension in *C. elegans*, several limitations exist. First, how the lysosomes or the v-ATPase/TORC1 complex senses the mitochondrial stress is not well addressed in the current work. To fill this gap, high-resolution imaging and systematic proteomic analysis of the changes in the content of endosomal and lysosomal vesicles in response to mitochondrial stress would be required in the future. Second, we did not manage to find a way to quantitate and compare the magnitude of the effects of v-ATPase/TORC1-mediated ATFS-1 translation versus the suppression of mitochondrial import as nonmutually exclusive mechanisms for UPR^mt activation. Third, despite that the decreased expression of ATFS-1 protein is sufficient to explain the reduced activation of UPR^mt upon v-ATPase/TORC1 inhibition in *C. elegans*, we cannot exclude that TORC1 or other enzymes, such as CBP-1 (Li et al., 2021), could still contribute to the UPR^mt through posttranslational modifications (e.g., phosphorylation, acetylation) of ATFS-1. Finally, it has been extensively reported that TORC1 signaling inhibition extends the lifespan in multiple animal models (Saxton and Sabatini, 2017; Shimobayashi and Hall, 2014). However, in our current study as well as in other

published work (Shpilka et al., 2021), TORC1 activity seems also to be essential for the mitochondrial stress response, which generally extends the lifespan as well. Thus, how these two intertwined processes coordinate to determine overall organismal health and lifespan still requires further investigation.

## Materials and methods

### *C. elegans* strains
The Bristol strain (N2) was used as the wild-type strain. *SJ4100 [zcIs13(hsp-6p::GFP)], MQ887 [isp-1(qm150)IV], MQ130 [clk-1(qm30) III], SJ4005 [zcIs4(hsp-4p::GFP)], CL2070 [dvIs70 (hsp-16.2p::GFP)], SJ4197 (zcIs39[dve-1p::dve-1::GFP]), OP506 (wgIs506 [xbp-1::TY1:: EGFP::3xFLAG + unc-119(+)]). OP675 (wgIs675 [atfs-1::TY1::EGFP:: 3xFLAG + unc-119(+)]), VC222 [raga-1(ok386)], HZ1683 [atg-2(bp576)], HZ1684 [atg-3(bp412)], HZ1686 [atg-7(bp411)], HZ1687 [atg-9(bp564)], RW12006 (stIs12006 [atfs-1::H1-wCherry + unc-119(+)]), RB967 [gcn-2(ok871)], and RB545 [pek-1(ok275)]* were obtained from the Caenorhabditis Genetics Center (CGC). Worms were routinely maintained at 20°C and fed with *Escherichia coli* (*E. coli*) OP50 (RRID:WB-STRAIN:WBStrain00041971) on Nematode Growth Medium (NGM) plates.

### RNA interference and drug treatment
*E. coli* strain HT115(DE3; RRID:WB-STRAIN:WBStrain00041080) was obtained from the CGC, transformed with the empty vector L4440, and used as the RNAi control. RNAi clones were obtained from either the Ahringer or Vidal library (Kamath et al., 2003;

Rual et al., 2004) and further verified by sequencing or qRT-PCR. The accession codes for *vha-1* RNAi clones were 10008-A6 (*vha-1_RNAi_1*) from the Vidal library and III-5A20 (*vha-1_RNAi_2*) from the Ahringer library. Double RNAi experiments were performed by mixing bacterial cultures normalized to their optical densities ($OD_{600}$) before seeding. For treatment of worms with compounds, stock solutions of Tunicamycin (Cat. T7765; Sigma-Aldrich), Concanamycin A (Cat. C9705; Sigma-Aldrich), Bafilomycin A1 (Cat. S1413; Selleckchem), or chloroquine (CQ, Cat. C6628; Sigma-Aldrich) were added to the NGM with final concentrations, as indicated in the figure legends, just before pouring the plates.

### UPRmt induction and imaging in *C. elegans*
For RNAi-induced UPRmt, RNAi bacteria were cultured in lysogeny broth (LB) medium containing 100 µg/ml ampicillin at 37°C overnight. The bacteria were then seeded onto NGM plates containing 2 mM IPTG and 25 mg/ml carbenicillin. L4/young adult worms were picked onto the RNAi bacteria-seeded plates and cultured at 20°C until their progenies reached the young adult stage. A total of 5–10 progenies were then randomly picked and aligned in 10 mM tetramisole (Cat. T1512; Sigma-Aldrich) droplets on NGM plates. Fluorescent images, with the same exposure time for each condition within each of the experiments, were acquired using a Nikon SMZ1000 microscope. All images are compared relatively only to each negative and positive controls in the same batch of the experiment. For compound-induced UPRmt, antimycin A (Cat. A8674; Sigma-Aldrich) with a final concentration of 2.5 µM or Dox (Cat. D9891; Sigma-Aldrich) with a final concentration of 30 µg/ml was added into the NGM just before pouring the plates. For imaging GFP-tagged ATFS-1, *atfs-1p::atfs-1::flag::gfp (OP675)* worms were mounted in 10 mM tetramisole (Cat. T1512; Sigma-Aldrich) on 2% agarose pads. The most proximal two intestinal cells in each worm were assessed with a Zeiss LSM 700 confocal microscope. At least 20 nuclei were analyzed for each condition. All images were acquired at room temperature with the software provided with the corresponding microscope.

### Lifespan experiments
Lifespan experiments were performed at 20°C as described previously (Houtkooper et al., 2013). Briefly, 80–100 worms were used per condition and scored every other day, and those that disappeared or exploded at the vulva were censored. Worms were transferred to fresh plates every week. All RNAi treatments for lifespan started at the maternal L4 stage.

### RNA extraction and RNA-seq analysis
For worm samples, synchronized worm eggs were seeded onto NGM plates and cultured at 20°C until the worms reached L4/young adult stage. Worms were then harvested with M9 buffer and snap-frozen in liquid nitrogen. For RNA extraction, 1 ml of TriPure Isolation Reagent (Cat. 11667165001; Roche) was added to each sample tube. The worms were then frozen with liquid nitrogen and thawed in a water bath quickly eight times to rupture cell membranes. Total RNA was then extracted using a column-based kit (Cat. 740955.250; Macherey-Nagel). All RNA-seq was performed by BGI with the BGISEQ-500 platform.

For RNA-seq results, the raw data were filtered by removing adaptor sequences, contamination, and low-quality (phred quality <20) reads. Qualified reads were then mapped to either the worm "*Caenorhabditis_elegans*.WBcel235.89" with STAR aligner version 2.6.0a. Reads were counted using htseq-count version 0.10.0 using these flags: -f bam -r pos -s no -m union -t exon -i gene_id. Differential expression of genes was calculated by Limma–Voom. The genes with a Benjamini–Hochberg adjusted P value <0.05 were defined as statistically significant. Genes whose expressions were significantly upregulated with $log_2FC > 0.393$ (the fold change for *hsp-6*, which encodes the UPRmt maker protein HSP-6 in *cco-1* RNAi condition and were then downregulated by more than 25% of the $log_2FC$ after *vha-1/-4/-16/-19* or *atfs-1* RNAi co-treatment, compared with the $log_2FC$ of *cco-1* RNAi condition, were considered as VHA- or ATFS-1-dependent. Functional clustering was conducted using the DAVID (Database for Annotation, Visualization, and Integrated Discovery) database (Huang et al., 2009). Heat-maps were generated by Morpheus (https://software.broadinstitute.org/morpheus). UpSet plot was generated by Intervene (https://asntech.shinyapps.io/intervene/; Khan and Mathelier, 2017).

### Quantitative RT-PCR (qRT-PCR)
Worms were harvested and total RNA was extracted as described above for RNA-seq. cDNA was synthesized using the Reverse Transcription Kit (Cat. 205314; Qiagen). qRT-PCR was conducted with the LightCycler 480 SYBR Green I Master kit (Cat. 04887352001; Roche). Primers used for qRT-PCR are listed in Table S2. Primers for worm *pmp-3* were used as the normalization control.

### Western blots
Proteins were extracted with Radio-immunoprecipitation Assay (RIPA) buffer supplied with protease and phosphatase inhibitors, as described previously (Houtkooper et al., 2013). For Western blots, the GFP-Flag tagged ATFS-1 in *atfs-1p::atfs-1::flag::gfp* or XBP-1 in *xbp-1p::xbp-1::flag::gfp* worms was detected by the anti-FLAG M2 antibody (Cat. F1804, 1:1,000, RRID:AB_262044; Sigma-Aldrich). The GFP tagged DVE-1 in *dve-1p::dve-1::gfp* worms were detected by the anti-GFP antibody (Cat. 2956, CST, 1:1,000, RRID:AB_1196615). P-RSKS-1 was detected using the phospho-*Drosophila* p70 S6 kinase (Thr398) antibody (Cat. 9209, 1:500, RRID:AB_2269804; CST), as validated in another study (Heintz et al., 2017). P-EIF-2α was detected using the P-EIF2α antibody (Cat. 3597, 1:500, RRID:AB_390740; CST), as validated in another study (Baker et al., 2012). Other antibodies used were: Tubulin (Cat. T5168, 1:2,000, RRID:AB_477579; Sigma-Aldrich), mCherry (Cat. 43590, 1:1,000, RRID:AB_2799246; CST), and HRP-labeled anti-rabbit (Cat. 7074, 1:5,000, RRID:AB_2099233; CST) and anti-mouse (Cat. 7076, 1:5,000, RRID:AB_330924; CST) secondary antibodies. Western blots were performed with either normal SDS-PAGE or 4–12 % gradient gels (Cat. M00654; GenScript).

## Polysome profiling

Polysome profiling was conducted as described previously (Gobet et al., 2020; Molenaars et al., 2020). Briefly, worms were collected and lysed in a polysome lysis buffer (100 mM KCl, 10 mM MgCl$_2$, 0.1% NP-40, 2 mM DTT, 0.5 mM Cycloheximide [Cat. S7418; Selleckchem], and RNasin Ribonuclease Inhibitor [Cat. N2611; Promega]) with a Dounce homogenizer. The samples were then centrifuged at 1,200 $g$ for 10 min to remove debris, and the supernatant was normalized using a *DC* protein assay (Cat. 5000112; Bio-Rad). Using the Gradient Master 108 programmable gradient pourer (Biocomp), sucrose gradients (17.5–50%) were generated in gradient buffer (20 mM Tris-HCl, 150 mM NaCl, 10 mM MgCl$_2$, 1 mM DTT, and 100 µg/ml Cycloheximide). Homogenized worm lysates containing 800 µg of total protein were then loaded onto the sucrose gradients and centrifuged at 32,000 rpm for 2.5 h in an SW40Ti rotor in a Beckman L7 ultracentrifuge (Beckman Coulter). After centrifugation, the gradients were fractionated and measured for RNA content (absorbance at 254 nm) using a Piston Gradient Fractionator (Biocomp) connected to a UV monitor (Bio-Rad). The polysomal fractions (P1–P6) were then collected and combined for RNA extraction using a column-based kit (Cat. 740955.250; Macherey-Nagel).

## Statistical analysis

No statistical methods were used to predetermine the sample size. Investigators were not blinded to allocation during experiments and outcome assessment. All experiments, except for the RNA-seq, were repeated at least twice, and similar results were acquired. All statistical analyses were performed using Graphpad Prism 8 software. Data distribution was assumed to be normal, but this was not formally tested. Differences between the two groups were assessed using two-tailed unpaired Student's *t* tests. Analysis of variance (ANOVA) followed by Tukey post-hoc test (one-way ANOVA for comparisons between groups, and two-way ANOVA for comparisons of magnitude of changes between different groups from different treatments or cell lines) was used when comparing more than two groups. Survival analyses were performed using the Kaplan–Meier method, and the significance of differences between survival curves was calculated using the log-rank (Mantel-Cox) method.

## Online supplemental material

Fig. S1 shows the impact of *vha-1*, *vha-4*, *vha-16*, or *vha-19* RNAi in different stress responses and gene expression in *C. elegans*. Fig. S2 shows the impact of mitochondrial stress, ER stress, and TORC1 signaling regulators in gene expression and stress responses. Fig. S3 shows the impact of ribosomal subunit RNAi in stress responses in *C. elegans*. Fig. S4 shows that autophagy-defective mutants have normal activation of the UPR$^{mt}$ in response to mitochondrial stress. Fig. S5 shows the key role of ribosomal subunits and VHA-1 in mitochondrial surveillance. Table S1 shows the RNA-seq results of worms fed with RNAi targeting v-ATPase subunits and/or *cco-1* RNAi. Table S2 shows the list of primers used for qRT-PCR in this study.

## Data availability

Original reagents are available upon request. The raw and processed sequencing datasets have been deposited in the NCBI Gene Expression Omnibus (GEO) database with the accession numbers: GSE179517.

## Acknowledgments

We thank the Caenorhabditis Genetics Center for providing the *C. elegans* strains. We thank all members of J. Auwerx and K. Schoonjans laboratories for helpful discussions.

This work was supported by grants from the Ecole Polytechnique Federale de Lausanne (EPFL), the European Research Council (ERC-AdG-787702), the Swiss National Science Foundation (SNSF 31003A_179435), and the GRL grant of the National Research Foundation of Korea (NRF 2017K1A1A2013124). T.Y. Li was supported by the "Human Frontier Science Program" (LT000731/2018-L). A.W. Gao was supported by the Accelerator prize given by the United Mitochondrial Disease Foundation (PF-19-0232). X. Li was supported by the China Scholarship Council (201906050019).

The authors declare no competing financial interests.

Author contributions: T.Y. Li and J. Auwerx conceived the project. T.Y. Li and A.W. Gao performed most of the experiments. T.Y. Li., A.W. Gao, X. Li, H. Li, and A. Lalou performed data analysis. A.W. Gao, Y.J. Liu, N. Neelagandan, and F. Naef contributed to the polysome profiling experiment. K. Schoonjans and J. Auwerx supervised and financed the study. T.Y. Li, A.W. Gao, and J. Auwerx wrote the manuscript with comments from all authors.

Submitted: 9 May 2022

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

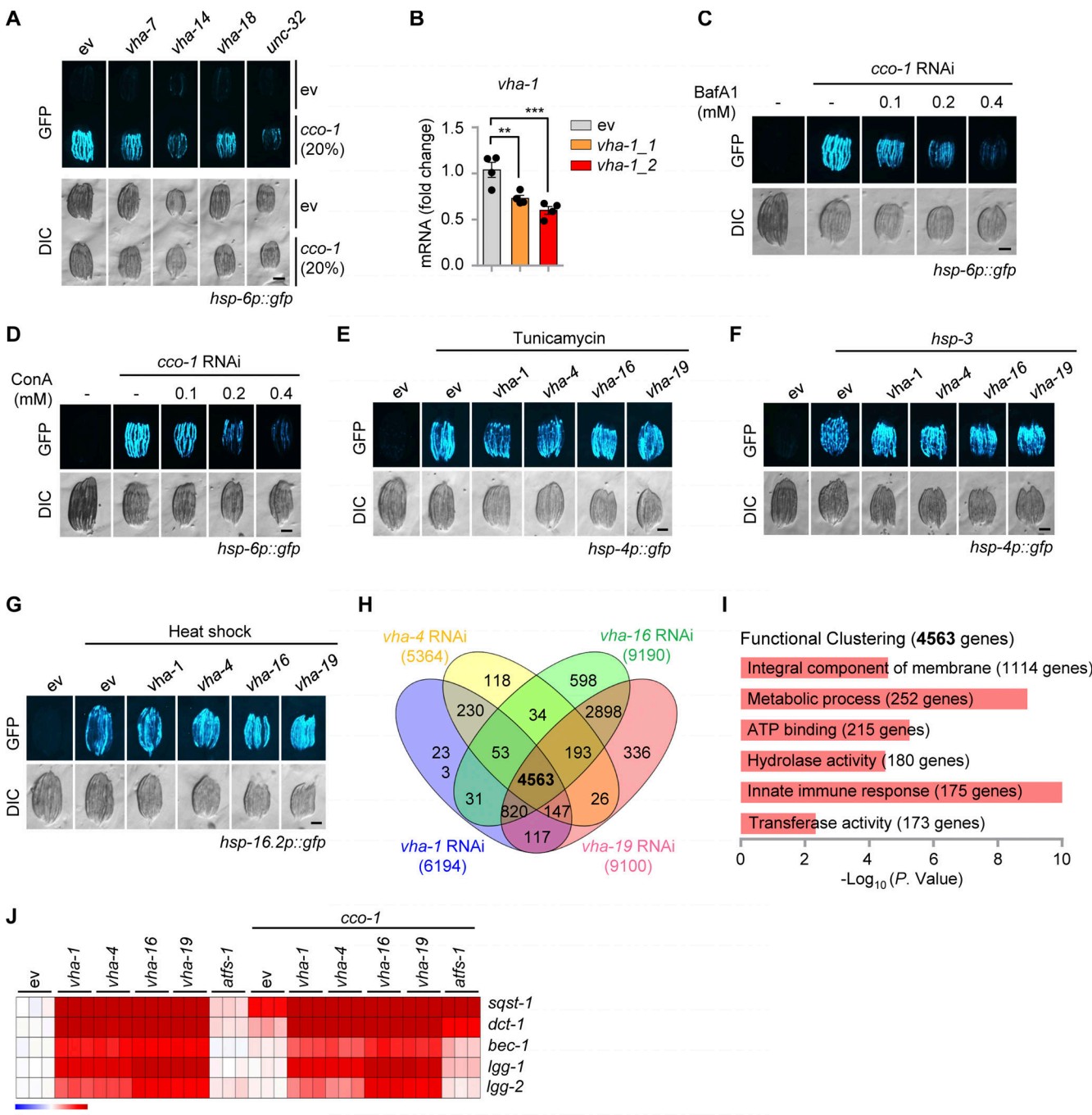

Figure S1. **Impact of *vha-1*, *vha-4*, *vha-16* or *vha-19* RNAi in different stress responses and gene expression in *C. elegans*. (A)** RNAi of multiple v-ATPase subunits attenuated UPR[mt] activation induced by *cco-1* RNAi in *hsp-6p::gfp* worms. For RNAi treatment, RNAi targeting v-ATPase subunits occupied 80%, *cco-1* RNAi occupied 20%. DIC, differential interference contrast image. **(B)** qRT-PCR analysis of *vha-1* mRNA (*n* = 4 biologically independent samples) in worms fed with control (ev), or *vha-1* RNAi obtained from either the Vidal (*vha-1*_RNAi_1) or Ahringer library (*vha-1*_RNAi_2). **(C and D)** The v-ATPase inhibitors, Bafilomycin A1 (BafA1; C) and Concanamycin A (ConA; D), dose-dependently suppressed the UPR[mt] induced by *mrps-5* or *cco-1* RNAi. **(E–G)** RNAi of *vha-1*, *vha-4*, *vha-16* or *vha-19* (25%) did not block the endoplasmic reticulum (ER) UPR (UPR[ER]) induced by tunicamycin (5 μg/ml; E) or *hsp-3* (40%) RNAi (F) or the cytosolic UPR (UPR[CYT])/heat shock response induced by heat shock at 30°C for 8 h (G). **(H)** Venn diagram of the differentially expressed genes in worms fed with RNAi as indicated. **(I)** Functional clustering of the 4,563 genes that were commonly affected by *vha-1*, *vha-4*, *vha-16*, and *vha-19* RNAi. **(J)** Heat map of the transcripts of representative mitophagy/autophagy-related genes in worms fed with RNAi as indicated; results are based on the RNA-seq dataset. Scale bars, 0.3 mm. Error bars denote SEM. Statistical analysis was performed by ANOVA followed by Tukey post-hoc test (**P < 0.01; ***P < 0.001).

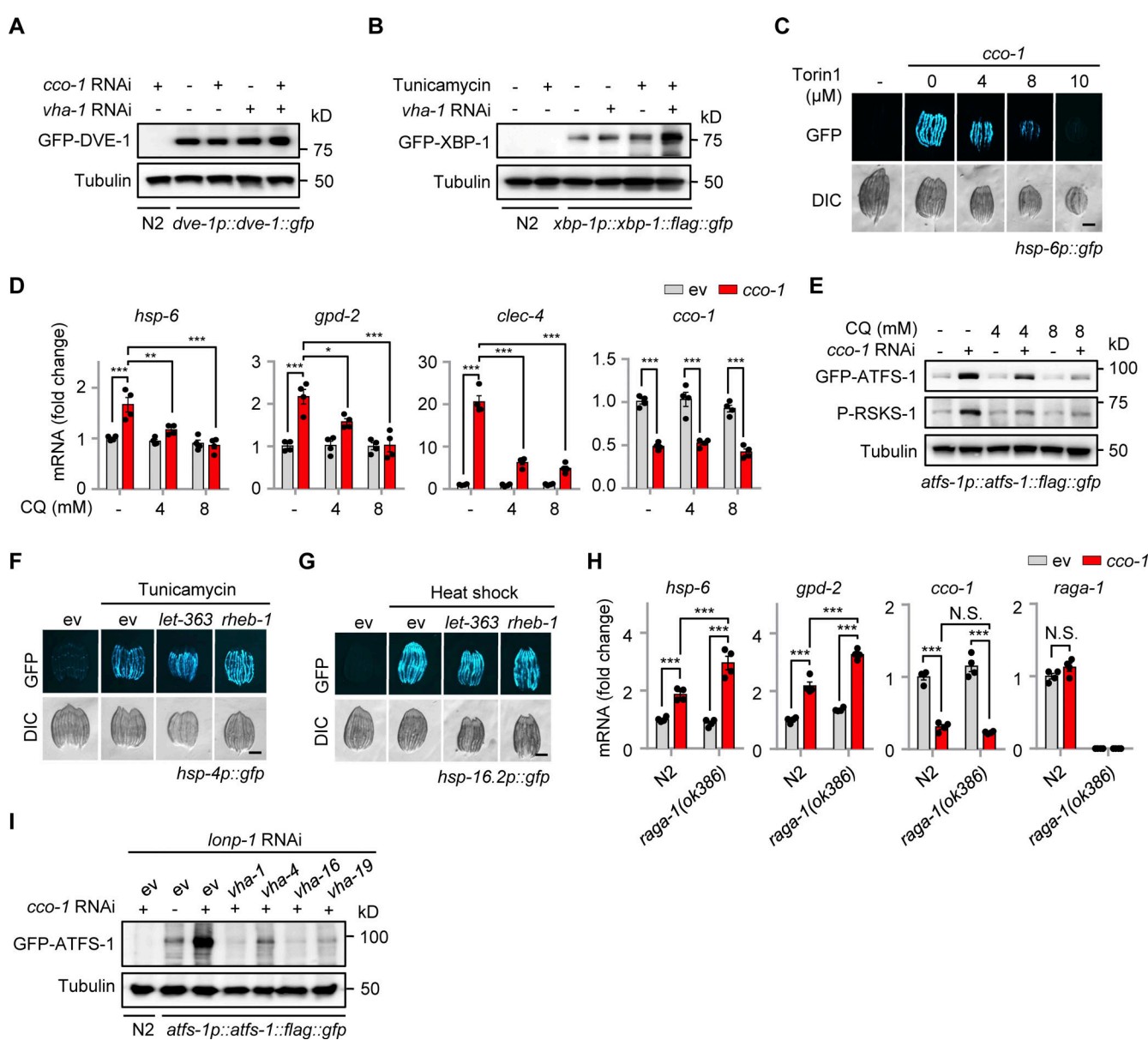

Figure S2. **Impact of mitochondrial stress, ER stress, and TORC1 signaling regulators in gene expression and stress responses. (A)** Western blots of *dve-1p::dve-1::gfp* worms fed with control, *cco-1* (50%) and/or *vha-1* (25%) RNAi. The wild-type (N2) worms fed with *cco-1* (50%) RNAi were used as a negative control. **(B)** Western blots of N2 or *xbp-1p::xbp-1::flag::gfp* worms fed with control or *vha-1* (25%) RNAi, with or without tunicamycin (2.5 µg/ml). **(C)** Torin1 suppressed *cco-1* RNAi-induced UPR^mt^ activation in a dose-dependent manner. *hsp-6p::gfp* worms were fed with control or *cco-1* (40%) RNAi, in combination with 4–10 µM Torin1. **(D and E)** qRT-PCR analysis of transcripts (*n* = 4 biologically independent samples; D) and Western blots (E) of *atfs-1p::atfs-1::flag::gfp* worms fed with control or *cco-1* (40%) RNAi, co-treated with or without 4 or 8 mM CQ. **(F and G)** RNAi of *let-363* and *rheb-1* (75%) did not block the UPR^ER^ induced by tunicamycin (5 µg/ml; F) or the UPR^CYT^ induced by heat shock at 30°C for 8 h (G). **(H)** qRT-PCR analysis of transcripts (*n* = 4 biologically independent samples) of wild-type or *raga-1(ok386)* worms fed with control or *cco-1* (50%) RNAi. **(I)** Western blots of N2 or *atfs-1p::atfs-1::flag::gfp* worms fed with control, *vha-1*, *vha-4*, *vha-16* or *vha-19* (25%) RNAi and/or *cco-1* (50%) RNAi, in the presence of *lonp-1* (25%) RNAi. Scale bars, 0.3 mm. Error bars denote SEM. Statistical analysis was performed by ANOVA followed by Tukey post-hoc test (*P < 0.05; **P < 0.01; ***P < 0.001). Source data are available for this figure: SourceData FS2.

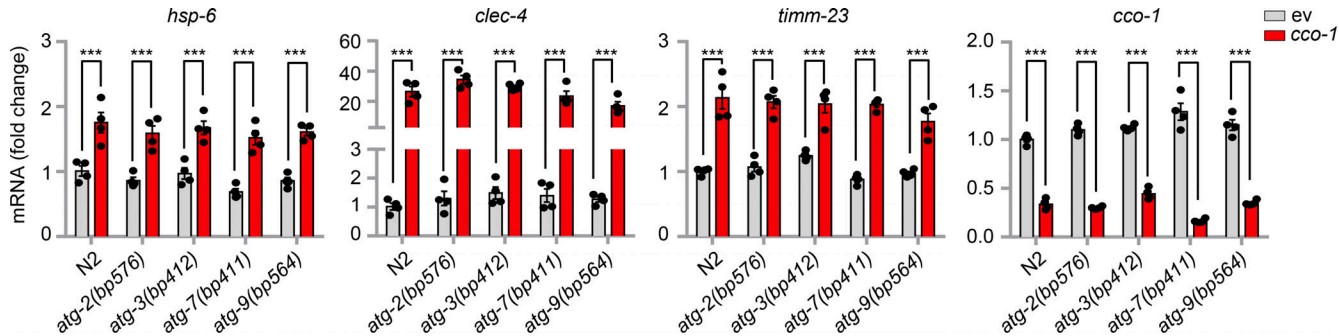

Figure S3.   **Impact of ribosomal subunit RNAi in stress responses in *C. elegans*. (A)** RNAi of ribosomal subunits strongly blocked the UPR$^{mt}$ activation induced by *mrps-5* RNAi. RNAi targeting *mrps-5* occupies 40%; RNAi targeting ribosomal subunits occupies 60%. **(B and C)** RNAi of ribosomal subunits only partially attenuated the UPR$^{ER}$ induced by tunicamycin (5 µg/ml; B) and did not affect the UPR$^{CYT}$ induced by heat shock at 30°C for 8 h (C). **(D)** qRT-PCR analysis of transcripts (*n* = 3 biologically independent samples) in the polysomal fractions of worms fed with control or *cco-1* (50%) RNAi, with or without or *vha-1* (25%) RNAi. **(E and F)** Total mRNA (*n* = 4 biologically independent samples; E) and polysomal mRNA (*n* = 3 biologically independent samples; F) level of *rsks-1* in worms fed with control or *cco-1* (50%) RNAi, with or without or *vha-1* (25%) RNAi. Scale bars, 0.3 mm. Error bars denote SEM. Statistical analysis was performed by ANOVA followed by Tukey post-hoc test (*P < 0.05; **P < 0.01; ***P < 0.001).

Figure S4.   **Autophagy defective mutants have normal activation of the UPR$^{mt}$ in response to mitochondrial stress.** qRT-PCR analysis of transcripts (*n* = 4 biologically independent samples) in wild-type (N2) or autophagy defect worm mutants fed with control or *cco-1* RNAi. Error bars denote SEM. Statistical analysis was performed by ANOVA followed by Tukey post-hoc test (***P < 0.001).

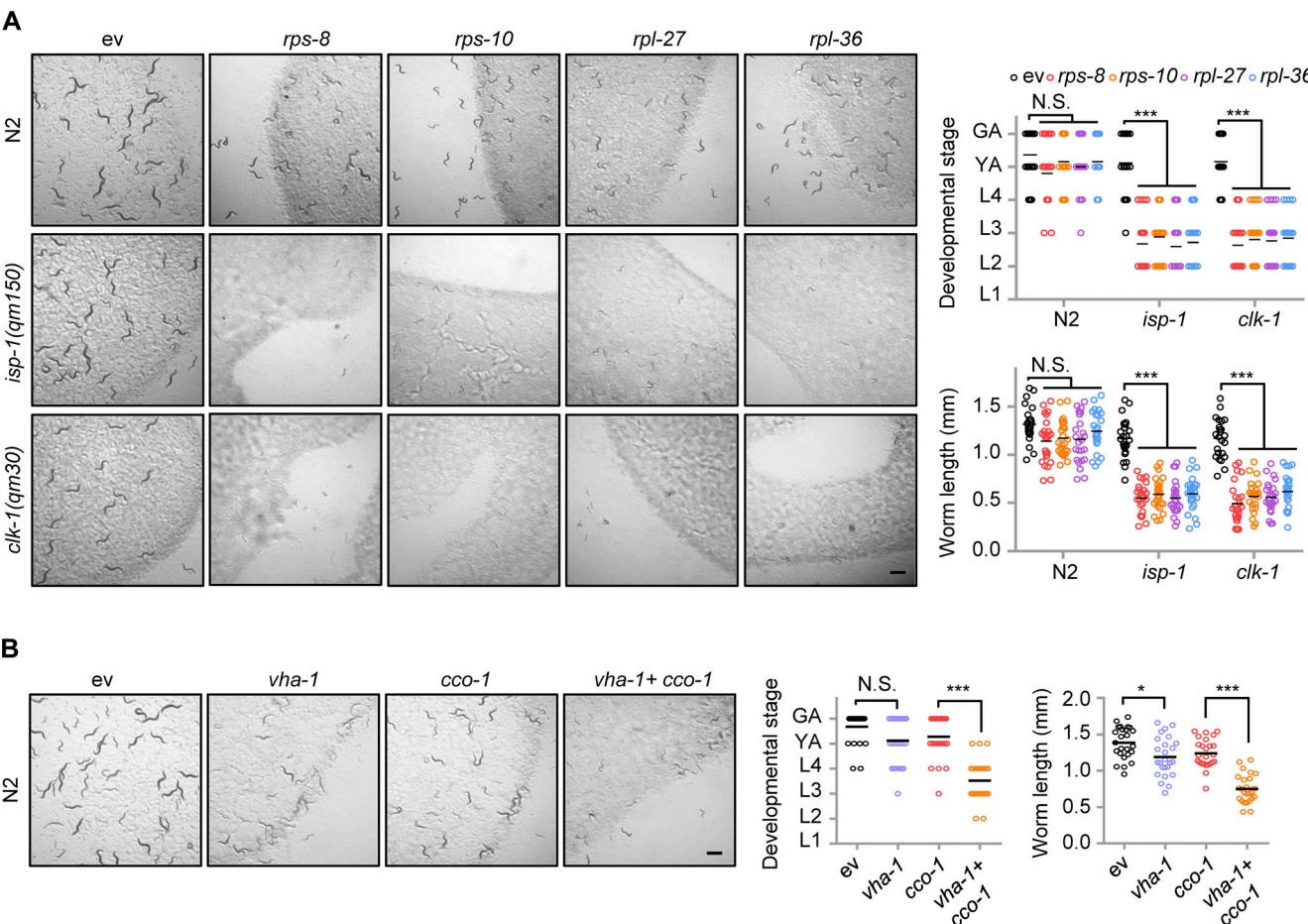

Figure S5. **A key role of ribosomal subunits and VHA-1 in mitochondrial surveillance. (A)** Representative brightfield pictures of wild-type (N2), *isp-1(qm150)* or *clk-1(qm30)* worms fed with control, *rps-8* (60%), *rps-10* (60%), *rpl-27* (60%), or *rpl-36* (60%) RNAi since the maternal L4 stage. Scale bar, 1 mm. The developmental stage (top right) and body length (bottom right) of the F1 progeny were quantified at day 4 after hatching (*n* = 25 worms for each condition). GA, gravid adult; L1-4, larval stages 1–4; YA, young adult. **(B)** Representative brightfield pictures of worms fed with control, *vha-1* (20%), and/or *cco-1* (50%) RNAi. The developmental stage and body length of the F1 progeny were quantified at day 4 after hatching (*n* = 25 worms for each condition). Error bars denote SEM. Statistical analysis was performed by ANOVA followed by Tukey post-hoc test (*P < 0.05; ***P < 0.001).

**Provided online are Table S1 and Table S2. Table S1 show the RNA-seq results of worms fed with RNAi targeting v-ATPase subunits and/or cco-1 RNAi. Table S2 show the list of primers used for qRT-PCR in this study.**

