## [Peer Review File · The Journal of Cell Biology]

V-ATPase/TORC1-mediated ATFS-1 translation directs mitochondrial UPR activation in *C. elegans*

Terytty Yang Li, Arwen Gao, Xiaoxu Li, Hao Li, Yasmine J. Liu, Amélia Lalou, Nagammal Neelagandan, Felix Naef, Kristina Schoonjans, and Johan Auwerx

Corresponding Author(s): Johan Auwerx, École Polytechnique Fédérale de Lausanne

Review Timeline:

Submission Date:	2022-05-09
Editorial Decision:	2022-06-15
Revision Received:	2022-09-22
Editorial Decision:	2022-10-04
Revision Received:	2022-10-11

Monitoring Editor: Jodi Nunnari

Scientific Editor: Andrea Marat

Transaction Report:

DOI: <https://doi.org/10.1083/jcb.202205045>

June 14, 2022

Re: JCB manuscript #202205045

Dr. Johan Auwerx
École Polytechnique Fédérale de Lausanne
Lab of Integrative Systems Physiology SV/IBI1/LISP/NCEM Station 15
Lausanne CH1015
Switzerland

Dear Johan,

Thank you for submitting your manuscript entitled "V-ATPase/TORC1-mediated ATFS-1 translation directs mitochondrial UPR activation in *C. elegans*". The manuscript was assessed by expert reviewers, whose comments are appended to this letter. We invite you to submit a revision if you can address the reviewers' key concerns, as outlined here.

You will see that the reviewers all appreciate that your paper provides interesting novel insight into mitochondrial stress signaling. We agree that your study opens up many exciting questions regarding the mechanistic links between the mitochondrial unfolded protein response, the V-ATPase, TORC1, and ATFS-1, and that fully understanding this new pathway will require follow up studies. Therefore, for resubmission to JCB experiments aimed at understanding the detailed mechanistic relationship between mitochondria, the V-ATPase, and TORC1 (notably reviewer 1 point 4, reviewer 2 point 1, reviewer 3 point 3) are not required, though this can be further discussed as noted by reviewer 2 minor points. In revising, we do expect you to experimentally address all comments pertaining to translational activation, examine if pH is a contributing mechanism, explain the differences with previous publications, as well as all other specific major and minor reviewer points.

GENERAL GUIDELINES:

Text limits: Character count for an Article is < 40,000, not including spaces. Count includes title page, abstract, introduction, results, discussion, and acknowledgments. Count does not include materials and methods, figure legends, references, tables, or supplemental legends.

Figures: Articles may have up to 10 main text figures. Figures must be prepared according to the policies outlined in our Instructions to Authors, under Data Presentation, <https://jcb.rupress.org/site/misc/ifora.xhtml>. All figures in accepted manuscripts will be screened prior to publication.

*****IMPORTANT:** It is JCB policy that if requested, original data images must be made available. Failure to provide original images upon request will result in unavoidable delays in publication. Please ensure that you have access to all original microscopy and blot data images before submitting your revision. ***

Supplemental information: There are strict limits on the allowable amount of supplemental data. Articles may have up to 5 supplemental figures. Up to 10 supplemental videos or flash animations are allowed. A summary of all supplemental material should appear at the end of the Materials and methods section.

Please note that JCB now requires authors to submit Source Data used to generate figures containing gels and Western blots with all revised manuscripts. This Source Data consists of fully uncropped and unprocessed images for each gel/blot displayed in the main and supplemental figures. Since your paper includes cropped gel and/or blot images, please be sure to provide one Source Data file for each figure that contains gels and/or blots along with your revised manuscript files. File names for Source Data figures should be alphanumeric without any spaces or special characters (i.e., SourceDataF#, where F# refers to the associated main figure number or SourceDataFS# for those associated with Supplementary figures). The lanes of the gels/blots should be labeled as they are in the associated figure, the place where cropping was applied should be marked (with a box), and molecular weight/size standards should be labeled wherever possible.

The typical timeframe for revisions is three to four months. While most universities and institutes have reopened labs and allowed researchers to begin working at nearly pre-pandemic levels, we at JCB realize that the lingering effects of the COVID-19 pandemic may still be impacting some aspects of your work, including the acquisition of equipment and reagents. Therefore, if you anticipate any difficulties in meeting this aforementioned revision time limit, please contact us and we can work with you to find an appropriate time frame for resubmission. Please note that papers are generally considered through only one revision cycle, so any revised manuscript will likely be either accepted or rejected.

Thank you for this interesting contribution to Journal of Cell Biology. You can contact us at the journal office with any questions, cellbio@rockefeller.edu or call (212) 327-8588.

Sincerely,

Jodi

Jodi Nunnari, Ph.D.
Editor-in-Chief

Andrea L. Marat, Ph.D.
Senior Scientific Editor

Journal of Cell Biology

Reviewer #1 (Comments to the Authors (Required)):

In this manuscript, Li et al. reported that a panel of vacuolar v-ATPase subunits is essential for the UPRmt activation in *C. elegans*. They showed that mitochondrial stress triggers the phosphorylation of RSKS-1, the *C. elegans* ortholog of mammalian S6 kinase, and a readout of mTORC1 activity. They further showed that mitochondrial stress increases TORC1 activity in a v-ATPase-dependent manner. Activated TORC1 leads to increased translation of the UPRmt transcription factor ATFS-1, thus resulting in the induction of UPRmt. Finally, the authors showed that v-ATPase RNAi also strongly abolished lifespan extension in worms with mild mitochondrial perturbations. The authors attempted to bring other cellular organelles (lysosomes and ribosomes) into the mitochondrial stress signaling. However, I have concerns regarding the mechanism by which activated TORC1 specifically increased translation of ATFS-1. It is unclear to explain how such specificity is determined. The data supporting the claim that the V-ATPase-TORC1-ATFS-1 pathway is weak and must be further substantiated.

Major points

1. A study published in PNAS (2020) showed that v-ATPase inhibitors suppressed the UPRmt in *drp-1* mutants, the mitochondrial fission regulator. RNAi of v-ATPase subunits *unc-32*, *vha-7*, *vha-14*, and *vha-18* strongly suppressed the UPRmt induction in *drp-1* mutants (PMID: 32737159). In this manuscript, the authors showed that only *vha-1*, *vha-4*, *vha-16*, and *vha-19* but not the other v-ATPase subunits are required for the UPRmt activation. The authors need to explain the discrepancy between these two groups of genes in terms of UPRmt regulation, such as the RNAi efficiency, subcellular localization, tissue specificity, etc.
2. Several previous studies have shown that inhibition of the TORC1 signaling pathway via RNAi against *rheb-1*, *let-363*, and *rsks-1* suppressed UPRmt activation. These studies also showed that *rheb-1* and *rsks-1* RNAi induced DVE-1::GFP expression. The authors need to explain the discrepancy of translational regulation between DVE-1 and ATFS-1, two main transcription factors for UPRmt activation.
3. In Figure 3A, the authors examined the level of phosphorylated RSKS-1 and concluded that v-ATPase RNAi suppressed increased phosphorylation of RSKS-1 in response to *cco-1* RNAi. The authors need to determine the total level of RSKS-1 for this comparison. The overall level of RSKS-1 may be altered upon v-ATPase RNAi.
4. The authors only showed that inhibition of the v-ATPase subunit suppressed the UPRmt activation. This suppression could be due to the v-ATPase activity or might be caused by indirect consequences of lysosomal dysfunction. Further, how is the v-ATPase activity regulated in response to mitochondrial stress conditions?

5. The data supporting that the translation of ATFS-1 is affected by the v-ATPase-TORC1-mediated translational regulation is relatively weak.

In Fig 4A and Fig S3B, the authors showed that ribosomal subunits RNAi remarkably suppressed *cco-1* RNAi-induced *hsp-6::gfp*, whereas tunicamycin-induced UPRER was only partially attenuated. It seems that tunicamycin-induced UPRER was also affected by ribosomal subunits RNAi, therefore these results are not very supportive of the conclusion.

ATFS-1 is a protein that needs to be processed by mitochondrial protein peptidases. From the western analysis in Fig 3A, the authors need to indicate which band represents the full-length of ATFS-1 and which one represents the mt-localized ATFS-1. It is also confusing to explain why mitochondrial *lonp-1* RNAi resulted in the accumulation of full-length ATFS-1, which is supposed to only affect the mitochondrial-localized ATFS-1.

In Fig 4D-F, it is confusing why the authors provided the results with the *atfs-1p::Histone-wCherry* transgenic strains. It seems that the histone protein exhibited similar patterns of translational regulation compared to the ATFS-1::GFP reporter, which is against their conclusion for specific translational regulation of ATFS-1 through the v-ATPase-TORC1 axis under mitochondrial stress conditions.

The imaging analysis of the CRISPR knock-in or a single copy insertion of ATFS-1::GFP transgenic worms is needed, especially for the subcellular localization of ATFS-1::GFP with or without *vha-1* RNAi under normal and mitochondrial stress conditions.

Minor point

The induction levels of *hsp-6p::gfp* by *cco-1*, and *mrps-5* RNAi in Fig 1D were not consistent with the results shown in Fig 1A and C.

Fig 1F, RNAi_2 showed strong suppression of UPRmt. RNAi efficiency needs to be tested.

Fig 2A, *vha-1* RNAi reduced the basal transcription level of *hsp-60* under normal conditions. Loss of v-ATPase might affect worm development, thus causing suppression of many developmental-related genes, including UPRmt target genes such as *hsp-6* and *atfs-1*.

Fig 4G and 4H, a single *vha-1* RNAi control is missing.

Fig 6B, *vha-16* and *vha-19* RNAi dramatically shorten lifespan, however, these results were not consistent in Fig 6C?

In Fig 3A, v-ATPase subunits RNAi suppressed the phosphorylation of RSKS-1 and the phosphorylation of eIF-2 α induced by mitochondrial stresses.

Reviewer #2 (Comments to the Authors (Required)):

In this manuscript, Li, Gao et al investigate how mitochondrial stress is signaled to the nucleus to mount the stress response called the mitochondrial unfolded protein response (UPR-mt). In a suppressor screen, they identify numerous components of the v-ATPase complex as required for the UPRmt. Interestingly, an ATFS-1::GFP fusion protein that is strongly upregulated during mito stress, is not induced when v-ATPase components are knocked down but this is not mirrored by their respective mRNA levels. LONP-1 is a negative ATFS-1 regulator and, as expected, its knockdown induces ATFS-1::GFP. However, mito stress has a stronger effect on ATFS-1 stabilisation and *lonp-1* is not sufficient for the induction of UPRmt target genes. So what mechanisms might be responsible for ATFS-1 accumulation?

Of note, the authors observe an activation of the TOR pathway during mito stress, which is likewise abrogated by *vha-1* RNAi. TOR activation is required for ATFS-1::GFP stabilisation.

Next focusing on mRNA translation, the authors find that knockdown of ribosomal subunits suppresses ATFS-1::GFP accumulation during mito stress, suggesting that ATFS-1 accumulates through increased *atfs-1* translation. Surprisingly, even histone H1 under the *atfs-1* promoter is upregulated by mito stress and this likewise is TOR, v-ATPase, and ribosomal subunit-dependent. Mito stress by *cco-1* RNAi enhances monosomes at the cost of polysomes while *atfs-1* mRNA is more polysome associated, suggesting active sorting.

While eIF2 α is phosphorylated under mito-stress, the UPRmt does not seem to depend on the ISR pathway.

Finally, the authors show that v-ATPase subunits are required for development and longevity of *isp-1* and *clk-1* mutants.

The paper tackles exciting and important questions, the experiments in the paper are well done and controlled. The paper raises some novel concepts but there are some gaps and it would be important to address some of them.

1. Does mito stress affect the v-ATPase? I wonder what is the mechanism by which mito-stress interacts with the v-ATPase

complex. Is the complex affected or does it purely act as an assembly point for TOR?

2. The authors suggest that v-ATPase depletion counters TOR activation. An alternative explanation would be an effect on ATPase function. I wonder if vacuolar pH might be affected and play a role?

3. Effects on general translation: mito stress seemed to reduce overall translation but this happened in the context of elevated TOR signaling. This seems contradictory but very interesting given the cited effect of TOR signaling of ATF-4. Certainly, it raises new questions around translation levels. What is overall translation as quantified by puromycin incorporation under the conditions used in the paper?

What is the effect of vha-1 RNAi alone on translation in a polysome profiling experiment?

4. Effects of atfs-1 translation: ATFS-1 abundance is increased by its mRNA translation even though polysomes are mildly depleted. To better understand this, it would be important to know the translational efficiency of the investigated mRNAs (polysome-associated vs total mRNA). I find it difficult to understand why H1 translation would be affected by translation when it is controlled under the atfs-1 promoter. Is the translational effect specific to atfs-1 or to all mRNAs? How could the promoter affect the protein's biosynthesis? Is the H1 construct perhaps cloned in a way that it puts H1 under the control of ATFS-1 translation initiation sequence, or is it trans-spliced? If so, this would be an opportunity to study atfs-1 translation.

5. To assess the processes contributing to ATFS-1 abundance, it would be important to consider its degradation as well.

Minor points:

As the authors suggest, it remains unclear how mito stress activates TOR signaling. I find it interesting that mito stress activates a "growth state" by triggering TOR. Activating TOR during stress seems to be a risky maneuver for cells. Moreover, I find it difficult to square the facts that TOR signaling is up during mito stress and that mito stress inhibits translation. It would be important to discuss this further, also in the light of the effect on survival: TOR inhibition extends survival but so does mito stress, which activates TOR. It would be important to further highlight these points in the discussion and to include a section discussing the limitations of the study.

Reviewer #3 (Comments to the Authors (Required)):

Here Li and colleagues searched for mediators of UPR_{mit} and found that RNAi of several vacuolar H⁺-ATPase components blocked expression of the UPR_{mit} reporter hsp-6p::gfp upon cco-1 RNAi in a dose-dependent manner, and supporting evidence includes knockdown of other mitochondrial stress inducers, UPR_{mit} activators, and knockdown of v-ATPase function, but separate from other UPR pathways and separate from mitophagy/autophagy. The expression of ATFS-1 itself seems to be regulated by v-ATPase, since cco-1 (RNAi) induces ATFS-1 protein but is blocked by knockdown of vha components. Linking to mTORC1, phosphorylation of RSKS-1 increases in a V-ATPase-dependent manner. V-ATPase regulation of ATFS-1 protein appears to be upstream of Lon protease function; therefore, the regulation of ATFS-1 protein must be at the translation level. Cytosolic ribosomal components are required for the ATFS-1 translation. The pathway also seems to be distinct from GCN-2, PEK-1, and EIF-2a kinase regulation.

Overall, the experiments are presented well and are logical, and describe a new, TORC1-specific pathway of UPR_{mit} activation that, surprisingly, has not been previously described. The only major shortcomings are the lack of information on the nature of the signal from the damaged mitochondria, and information on whether translation is specifically and only of ATFS-1, or if there is a suite of proteins whose translation is activated in this manner.

Major questions/suggestions:

1. Why do the different components not all have the same effect - is that an RNAi artifact (some clones don't work as well?), or some functional difference between the subunits? While I first thought the former must be true, the fact that the RNA-seq identifies expression differences that seem specific to the subunit knocked down is surprising. I realize that the 4-way Venn is appealing, but perhaps the proportional Venn diagram (using boxes) might better convey the fact that the majority of the cco-1-induced v-ATPase-dependent genes (325) are in fact shared - and thus suggests that the differences might be minor and due to RNAi efficacy differences. (The PCA also emphasizes differences, but might be unavoidable.)

2. Figure 6 - I understand that isp-1 and clk-1 are mutants and therefore could be treated with vha RNAis to check growth, but perhaps images of the double RNAi of cco-1 and vha could be shown here as well in Fig. 6A (i.e., like Fig. 1A but in N2 rather than hsp-6p::gfp) - easy to add these photos since the lifespans are already shown.

3. The model as shown makes sense, but what is the signal from stressed mitochondria to RHEB and TORC1 to result in such a specific translation of ATFS-1? The authors acknowledge this gap, but do the authors think there is a specific peptide that is transported to the lysosomes that somehow activates RHEB, or is it a general response to increased vacuolar occupancy upon increased unfolded proteins upon mitochondrial stress? Could they test the contact site model?

4. Similarly, the specific activation in translation of ATFS-1 seems solid. Are there other proteins that are also increased by translation activation along with ATFS-1 as a result of this TORC1-mediated translation activation?

5. Is there a way to quantitate and compare the magnitudes of the effects of TORC1-mediated ATFS-1 translation vs ATFS-1 nuclear translocation into the nucleus/Lon protease mechanism, so we can understand their relative importance?

Minor (writing suggestions)

1. "Despite that some evidence implicated the involvement of TORC1 components" - revise to "Despite some evidence that implicated" or "Despite the fact that some evidence implicated"
2. "majorly" would read better as "primarily"
3. "mechanism on how TORC1"  "mechanism of how" (this appears several times in the paper)
4. "little is known on how"  "little is known about how" or "little is known of how"
5. "Furthermore, whether the communications between" communication (does not need to be written as plural)
6. It might be better to group all of the initial TOR pathway results together (RSKS-1, let-363, rheb-1, Torin1), rather than breaking them up with the lonp-1 results in between, if there is no reason to split them (I can't see an obvious reason why Lon is put before the other TOR components - if there is a reason they are separated, make it clearer to the reader).

Reviewer #1:

In this manuscript, Li et al. reported that a panel of vacuolar v-ATPase subunits is essential for the UPR^{mt} activation in *C. elegans*. They showed that mitochondrial stress triggers the phosphorylation of RSKS-1, the *C. elegans* ortholog of mammalian S6 kinase, and a readout of mTORC1 activity. They further showed that mitochondrial stress increases TORC1 activity in a v-ATPase-dependent manner. Activated TORC1 leads to increased translation of the UPR^{mt} transcription factor ATFS-1, thus resulting in the induction of UPR^{mt}. Finally, the authors showed that v-ATPase RNAi also strongly abolished lifespan extension in worms with mild mitochondrial perturbations. The authors attempted to bring other cellular organelles (lysosomes and ribosomes) into the mitochondrial stress signaling. However, I have concerns regarding the mechanism by which activated TORC1 specifically increased translation of ATFS-1. It is unclear to explain how such specificity is determined. The data supporting the claim that the v-ATPase-TORC1-ATFS-1 pathway is weak and must be further substantiated.

Answer: We thank the reviewer for the constructive comments regarding the mechanistic aspect of our manuscript. In the revised manuscript, we have now provided detailed explanations as well as new results to further strengthen our main conclusions. Below please find our point-by-point responses to each of your comments.

Major points:

1. A study published in PNAS (2020) showed that v-ATPase inhibitors suppressed the UPR^{mt} in *drp-1* mutants, the mitochondrial fission regulator. RNAi of v-ATPase subunits *unc-32*, *vha-7*, *vha-14*, and *vha-18* strongly suppressed the UPR^{mt} induction in *drp-1* mutants (PMID: 32737159). In this manuscript, the authors showed that only *vha-1*, *vha-4*, *vha-16*, and *vha-19* but not the other v-ATPase subunits are required for the UPR^{mt} activation. The authors need to explain the discrepancy between these two groups of genes in terms of UPR^{mt} regulation, such as the RNAi efficiency, subcellular localization, tissue specificity, etc.

Answer: We thank the reviewer to point out the discrepancy between our findings and those reported by Wei and Ruvkun, 2020, PNAS. In the PNAS study which was performed in *drp-1* mutant worms, the worms were fed with RNAi of *unc-32*, *vha-7*, *vha-14*, and *vha-18* at 100%. In contrast, in our study, the v-ATPase RNAis were used at 60% of the total RNAi load, since the *cco-1* RNAi occupied the remaining 40% (Fig. 1A). Therefore, it is possible that the RNAi efficiency explains this discrepancy. Indeed, while worms fed with RNAi of *vha-1*, *vha-4*, *vha-16*, and *vha-19* appeared smaller in size, worms fed with RNAi of *unc-32*, *vha-7*, *vha-14*, and *vha-18* were almost not different in size (Fig. 1A), suggesting a lower RNAi efficiency. Additionally, *cco-1* (40%) RNAi may induce the UPR^{mt} to a more pronounced extent than that observed in *drp-1* mutant worms. To validate this assumption, we reduced the *cco-1* RNAi occupancy to 20%; allowing to increase the RNAi of *unc-32*, *vha-7*, *vha-14*, and *vha-18* to 80%. As shown in the new Figure S1A, we found that 80% RNAi of *unc-32*, *vha-7*, *vha-14*, and *vha-18* indeed attenuated the UPR^{mt} activation by *cco-1* (20%) RNAi to some extent and also reduced the worm size. Furthermore, the effect of *vha-14* and *unc-32* RNAi on UPR^{mt} suppression was more potent than that of *vha-7* and *vha-18* RNAi (new Fig. S1A), in line with the results in the *drp-1* mutants (Wei and Ruvkun, 2020, PNAS, Suppl. Fig. 4C). These results have now also been included in the revised manuscript (Fig. S1A, Page 4, Line 29).

2. Several previous studies have shown that inhibition of the TORC1 signaling pathway via RNAi against *rheb-1*, *let-363*, and *rsk-1* suppressed UPR^{mt} activation. These studies also showed that *rheb-1* and *rsk-1* RNAi induced DVE-1::GFP expression. The authors need to explain the discrepancy of translational regulation between DVE-1 and ATFS-1, two main transcription factors for UPR^{mt} activation.

Answer: Indeed, several previous studies have shown that inhibition of components in the TORC1 signaling suppressed UPR^{mt} activation (PMID: 17925224, PMID: 22719267, PMID: 33473112). However, rather than showing that *rheb-1* and *rsk-1* RNAi induced DVE-1::GFP expression, these studies actually suggested that "inactivation of either Rheb homolog or *C. elegans* TOR blocks *hsp-60pr::gfp* expression in stressed worms but does not interfere with *dve-1* or *ubl-5*. Rather, *rheb-1* inactivation promotes nuclear redistribution of DVE-1, induction of *ubl-5*, and complex formation with DVE-1" (PMID: 17925224). Consistently, our results indicated that while the ATFS-1 protein is regulated by mitochondrial stress in a v-ATPase/TORC1-dependent fashion (updated Fig. 3 A, D and F), the total DVE-1::GFP expression is neither affected by mitochondrial stress nor by v-ATPase RNAi (Fig. S2A). To further clarify the discrepancy of translational regulation between DVE-1 and ATFS-1, we further checked the total and polysomal-specific mRNA levels of *dve-1* with or without *vha-1* RNAi in the absence or presence of mitochondrial stress. The new results showed that both the total and polysomal-specific mRNA level of *dve-1* is not affected by mitochondrial stress, and was conversely increased upon *vha-1* RNAi (Table S1 and new Fig. S3D), suggesting that different from ATFS-1, the translational regulation of DVE-1 is independent of mTORC1 signaling. These new results have now been included in the revised manuscript (Fig. S3D, Page 9, Line 33).

3. In Figure 3A, the authors examined the level of phosphorylated RSKS-1 and concluded that v-ATPase RNAi suppressed increased phosphorylation of RSKS-1 in response to *cco-1* RNAi. The authors need to determine the total level of RSKS-1 for this comparison. The overall level of RSKS-1 may be altered upon v-ATPase RNAi.

Answer: We thank the reviewer for pointing this out. Unfortunately, there is no validated total RSKS-1 antibody available in the literature or typical commercial market. To address this concern in an alternative way, we have now measured both the total and polysomal mRNA levels of *rsk-1* in worms fed with control or *vha-1* RNAi, in the

absence or presence of mitochondrial stress (*cco-1* RNAi). We found that neither *vha-1* RNAi nor mitochondrial stress affected the total and polysomal mRNA levels of *rsk-1* (new Fig. S3, E and F). Consistently, the polysomal mRNA level of *rsk-1* is also not affected by *mmps-5* RNAi in another study (PMID: 32084377). Together, these results suggest that the transcription and translation of *rsk-1* was likely not affected upon v-ATPase or mitochondrial stress. These results have now also been included in the revised manuscript (Fig. S3, E and F; Page 10, Line 3).

4. *The authors only showed that inhibition of the v-ATPase subunit suppressed the UPR^{mt} activation. This suppression could be due to the v-ATPase activity or might be caused by indirect consequences of lysosomal dysfunction. Further, how is the v-ATPase activity regulated in response to mitochondrial stress conditions?*

Answer: Actually, in our model, the overall lysosomal function likely also contributes to the UPR^{mt} activation process, since TORC1 activation requires functional lysosomes as well (PMID: 31768005; PMID: 25567907; PMID: 32285908). To further validate this point, we have used another lysosomal acidification inhibitor chloroquine (CQ), and found that CQ inhibited UPR^{mt} in a dose-dependent manner (new Fig. 1H and S2D), suppressed TORC1 activity as well as the accumulation of ATFS-1 upon mitochondrial stress (new Fig. S2E). These results support our model that the intact lysosomal function and lysosomal pH are required for TORC1 and UPR^{mt} activation. We feel that the investigation of the detailed mechanistic relationship between v-ATPase activity, lysosomal dysfunction and mitochondrial stress is out the scope of the current study and will be addressed in further work. The new data with CQ have now also been included and discussed in the revised manuscript (Fig. 1H, S2D and S2E; Page 5, Line 11; Page 7, Line 31).

5. *The data supporting that the translation of ATFS-1 is affected by the v-ATPase-TORC1-mediated translational regulation is relatively weak. In Fig 4A and Fig S3B, the authors showed that ribosomal subunits RNAi remarkably suppressed *cco-1* RNAi-induced *hsp-6::gfp*, whereas tunicamycin-induced UPR^{ER} was only partially attenuated. It seems that tunicamycin-induced UPR^{ER} was also affected by ribosomal subunits RNAi, therefore these results are not very supportive of the conclusion.*

Answer: While we agree with the reviewer that tunicamycin-induced UPR^{ER} was partially affected by RNAi of the ribosomal subunits, we would also like to point out that the effect of RNAi of the ribosomal subunits in suppressing *cco-1* or *mmps-5* RNAi-induced HSP-6-GFP expression is far more pronounced than that on tunicamycin-induced UPR^{ER} (Fig. 4A, S3A and S3B), in line with a previous study (PMID: 23516373). Additionally, we have also shown that the UPR^{CYT}/heat shock response is not affected by RNAi of the ribosomal subunit RNAi (Fig. S3C). We thereby consider that these results are still informative and generally support our conclusion.

6. *ATFS-1 is a protein that needs to be processed by mitochondrial protein peptidases. From the western analysis in Fig 3A, the authors need to indicate which band represents the full-length of ATFS-1 and which one represents the mt-localized ATFS-1. It is also confusing to explain why mitochondrial *lonp-1* RNAi resulted in the accumulation of full-length ATFS-1, which is supposed to only affect the mitochondrial-localized ATFS-1.*

Answer: Due to the fact that the full-length ATFS-1 and the mt-localized ATFS-1 only have ~5 kDa difference in molecular weight (PMID: 22700657), it is not easy to differentiate between them with normal SDS-PAGE gels, especially with the C-terminal Flag-GFP tag added to the ATFS-1 protein in the *atfs-1p::atfs-1::flag::gfp* strain we utilized. To improve the separation of ATFS-1 proteins in worm samples fed with *lonp-1* and/or *cco-1* RNAi, we used the precast 4-12 % gradient gel (Cat. M00654, GenScript) and the results now show that *lonp-1* RNAi indeed only increases the accumulation of the lower (mt-localized ATFS-1) band of ATFS-1, while *cco-1* RNAi leads to the accumulation of both the lower (mt-localized ATFS-1) and upper (full length) bands of ATFS-1 (updated Fig. 3H). This new result has now been included and better discussed in the revised manuscript (Fig. 3H, Page 8, Line 18).

7. *In Fig 4D-F, it is confusing why the authors provided the results with the *atfs-1p::Histone-wCherry* transgenic strains. It seems that the histone protein exhibited similar patterns of translational regulation compared to the ATFS-1::GFP reporter, which is against their conclusion for specific translational regulation of ATFS-1 through the v-ATPase-TORC1 axis under mitochondrial stress conditions.*

Answer: The *atfs-1p::Histone-wCherry* transgenic strain was constructed such that the expression of the Histone-mCherry reporter protein is under the strict control of the upstream intergenic sequences (including both the promoter and 5'-UTR regions) of *atfs-1* (PMID: 22508763). Therefore, similar to the ATF4 translation sensors constructed in mammalian systems (PMID: 30088945), the expression/translation of this Histone-mCherry is controlled in a similar fashion as that of the endogenous ATFS-1 protein. In addition, since the translated Histone-mCherry is not degraded by LONP-1 or other ATFS-1-specific-targeting enzymes, the *atfs-1p::Histone-wCherry* transgenic strain is therefore an ideal system to study ATFS-1 translation regulation, independent of its degradation. The data with the Histone-mCherry protein (controlled by the 5'-UTR of *atfs-1*) that exhibited a similar expression pattern as the ATFS-1::GFP reporter (Fig. 4D-4F), hence fully support our current model that the increased expression of ATFS-1 protein under mitochondrial stress is driven by the v-ATPase-TORC1-mediated translational regulatory mechanism, independent of the degradation of ATFS-1. We have now elaborated more on the specifics of the *atfs-1p::Histone-wCherry* transgenic strain in the revised manuscript to clarify this point (Page 9, Line 11).

8. *The imaging analysis of the CRISPR knock-in or a single copy insertion of ATFS-1::GFP transgenic worms is*

needed, especially for the subcellular localization of ATFS-1::GFP with or without *vha-1* RNAi under normal and mitochondrial stress conditions.

Answer: As suggested, by using the *atfs-1p::atfs-1::flag::gfp* strain (OP675: <https://cgc.umn.edu/strain/OP675>) with a EGFP::3xFLAG tag inserted in frame at the C-terminus of the genomic locus of *atfs-1*, we found that mitochondrial stress (*cco-1* RNAi) leads to the nuclear accumulation of ATFS-1::GFP (as reported previously: PMID: 22700657), which was not seen in worms with the co-treatment of *vha-1*, *vha-4*, *vha-16* and *vha-19* RNAi (new Fig. 2I; Page 6, Line 13 in the revised manuscript). These results support our model that the v-ATPase subunits are required for the increased expression and the subsequent nuclear-localization of ATFS-1 for UPR^{mt} activation.

Minor points:

1. The induction levels of *hsp-6p::gfp* by *cco-1*, and *mrps-5* RNAi in Fig 1D were not consistent with the results shown in Fig 1A and C.

Answer: The induction level of *hsp-6p::gfp* may indeed show some inevitable variations between different batches. Therefore, we have always included a negative (without mitochondrial stress) and positive control (mitochondrial stressed) in each experimental batch; all images are hence compared relatively only to each negative and positive controls in the same batch. We have now clarified this in the Methods section (Page 16, Line 1). Meanwhile, to avoid any further confusions, we have also replaced the original Fig. 1D GFP image with a longer exposure-time photo (the overall exposure time for each RNAi treatment condition within this batch remains the same).

2. Fig 1F, RNAi_2 showed strong suppression of UPR^{mt}. RNAi efficiency needs to be tested.

Answer: By checking the knockdown efficiency of *vha-1* RNAi_1 and RNAi_2, we found that the strong suppression of *vha-1* RNAi_2 is indeed linked with better knockdown efficiency, as compared to that of *vha-1* RNAi_1 (New Fig. S1B). This is now mentioned on Page 5, line 5 in the revised manuscript.

3. Fig 2A, *vha-1* RNAi reduced the basal transcription level of *hsp-60* under normal conditions. Loss of v-ATPase might affect worm development, thus causing suppression of many developmental-related genes, including UPR^{mt} target genes such as *hsp-6* and *atfs-1*.

Answer: This is a good point. We have now added the discussion on the role of v-ATPase in maintaining basal UPR^{mt} activity in the revised manuscript (Page 13, Line 8). We would also like to point out that the distinction between basal and stress conditions is somehow artificial, especially considering that organisms and cells are constantly exposed to multiple cues, and different wild *C. elegans* strains differ with respect to the level of UPR^{mt} activation under basal conditions (PMID: 29120414). Additionally, delay of the worm development does not necessarily associate with the suppression of UPR^{mt}. In fact, most of the mitochondrial stress inducers delay worm development (PMID: 21215371; PMID: 23698443).

4. Fig 4G and 4H, a single *vha-1* RNAi control is missing.

Answer: The primary objective of this experiment was to study the effect of *vha-1* RNAi on the translation of *atfs-1* during mitochondrial stress, we therefore did not include the *vha-1* RNAi data initially. In the revised manuscript, we have now also included the single *vha-1* RNAi condition data (updated Fig. 4G and 4H).

5. Fig 6B, *vha-16* and *vha-19* RNAi dramatically shorten lifespan, however, these results were not consistent in Fig 6C?

Answer: We went through our original lifespan scoring sheets and found that we inadvertently failed to include the scored dead worms before Day 10 in Fig. 6C. We have now fixed this error in the revised manuscript (updated Fig. 6C), which validates that *vha-16* and *vha-19* RNAi robustly shorten lifespan.

6. In Fig 3A, v-ATPase subunits RNAi suppressed the phosphorylation of RSKS-1 and the phosphorylation of eIF-2 α induced by mitochondrial stresses.

Answer: In contrast to the dramatic changes of RSKS-1 phosphorylation, the phosphorylation of EIF-2 α is only modestly decreased (upon *vha-1*, *vha-4* and *vha-19* RNAi), or unaffected (upon *vha-16* RNAi). Accordingly, we have described this result more accurately in the revised manuscript (Page 7, Line 1).

Reviewer #2:

In this manuscript, Li, Gao et al investigate how mitochondrial stress is signaled to the nucleus to mount the stress response called the mitochondrial unfolded protein response (UPR^{mt}). In a suppressor screen, they identify numerous components of the v-ATPase complex as required for the UPR^{mt}. Interestingly, an ATFS-1::GFP fusion protein that is strongly upregulated during mito stress, is not induced when v-ATPase components are knocked down but this is not mirrored by their respective mRNA levels. LONP-1 is a negative ATFS-1 regulator and, as expected, its knockdown induces ATFS-1::GFP. However, mito stress has a stronger effect on ATFS-1 stabilization and lonp-1 is not sufficient for the induction of UPR^{mt} target genes. So what mechanisms might be responsible for ATFS-1 accumulation? Of note, the authors observe an activation of the TOR pathway during mito stress, which is likewise abrogated by vha-1 RNAi. TOR activation is required for ATFS-1::GFP stabilization. Next focusing on mRNA translation, the authors find that knockdown of ribosomal subunits suppresses ATFS-1::GFP accumulation during mito stress, suggesting that ATFS-1 accumulates through increased atfs-1 translation. Surprisingly, even histone H1 under the atfs-1 promoter is upregulated by mito stress and this likewise is TOR, v-ATPase, and ribosomal subunit-dependent. Mito stress by cco-1 RNAi enhances monosomes at the cost of polysomes while atfs-1 mRNA is more polysome associated, suggesting active sorting. While eIF2a is phosphorylated under mito-stress, the UPR^{mt} does not seem to depend on the ISR pathway. Finally, the authors show that v-ATPase subunits are required for development and longevity of isp-1 and clk-1 mutants. The paper tackles exciting and important questions, the experiments in the paper are well done and controlled. The paper raises some novel concepts but there are some gaps and it would be important to address some of them.

Answer: We thank the reviewer for finding that our work tackles exciting and important questions, and is of high quality. We would also like to express our sincere appreciation for the constructive comments/suggestions. Below please find our point-by-point responses to each of your comments.

1. Does mito stress affect the v-ATPase? I wonder what is the mechanism by which mito-stress interacts with the v-ATPase complex. Is the complex affected or does it purely act as an assembly point for TOR?

Answer: Although defining the exact mechanism how mitochondrial stress impacts on the v-ATPase/mTORC1 complex is interesting, we consider it beyond the scope of the current study, which aimed to reveal the key role of v-ATPase/mTORC1-mediated ATFS-1 translation in UPR^{mt} activation. Nevertheless, we have used the lysosomal acidification inhibitor chloroquine (CQ) and found that the maintenance of lysosomal pH is essential for TORC1 activation and UPR^{mt} activation (as specified in the next Point #2), indicating that the role of v-ATPase complex in lysosomal acidification probably also contributes to the sensing of mitochondrial stress. The related results (new Fig. 1H, S2D and S2E) and discussions (Page 5, Line 11; Page 7, Line 31) are now included in the revised manuscript.

2. The authors suggest that v-ATPase depletion counters TOR activation. An alternative explanation would be an effect on ATPase function. I wonder if vacuolar pH might be affected and play a role?

Answer: Actually, in our model and as explained in the discussion section, the overall lysosomal function likely also contributes to the UPR^{mt} activation process, since TORC1 activation requires functional lysosomes as well (PMID: 31768005; PMID: 25567907; PMID: 32285908). To further validate this point, we used another lysosomal acidification inhibitor chloroquine (CQ), and found that CQ inhibited UPR^{mt} in a dose-dependent manner (new Fig. 1H and S2D), suppressed TORC1 activity as well as the accumulation of ATFS-1 upon mitochondrial stress (new Fig. S2E). These results suggest that vacuolar pH and intact lysosomal function are required for TORC1 and UPR^{mt} activation.

3. Effects on general translation: mito stress seemed to reduce allover translation but this happened in the context of elevated TOR signaling. This seems contradictory but very interesting given the cited effect of TOR signaling of ATF-4. Certainly, it raises new questions around translation levels. What is allover translation as quantified by puromycin incorporation under the conditions used in the paper?

Answer: We thank the reviewer for the insightful comment. As suggested, we performed the puromycin incorporation assay in worms fed with control or vha-1 RNAi, with or without mitochondrial stress (cco-1 RNAi), and the results are as shown below (Fig. R1). On one hand, we found that mitochondrial stress (cco-1 RNAi) overall reduces translation, in line with the ribosomal profiling data (Updated Fig. 4G). However, for some proteins at specific regions (e.g., region #3), more puromycin incorporation was also found in mitochondrial stressed worms as compared to that in control condition. On the other hand, RNAi of vha-1 led to reduced puromycin incorporation for some high molecular weight proteins (e.g., region #1), but also increased puromycin incorporation for many low molecular weight proteins (e.g., region #2 - #4). Therefore, based on this puromycin incorporation results, it is rather difficult to draw a solid conclusion on how mitochondrial stress or vha-1 RNAi affect the overall protein translation. Moreover, it has been reported that the puromycin labeling assay may not be suitable for measuring the rates of overall protein synthesis under conditions of energy starvation (PMID: 29348556), which probably occurs during mitochondrial stress as well as vha-1 RNAi condition. Furthermore, according to the ribosomal profiling data (Updated Fig. 4G), vha-1 RNAi alone generally reduced translation. Due to the complexity of the information acquired from the puromycin labeling data, we opted not to include this result in the revised manuscript.

Figure R1. Western blots of Day 1 wild-type worms fed with control, *vha-1* (25%) and/or *cco-1* (50%) RNAi, with or without puromycin labeling (final puromycin concentration: 0.5 mg/mL) for 3 h, following a protocol as described previously (PMID: 33723245). The primary antibodies used were: anti-Puromycin (Cat. MABE343, Merck, 1:1000) and anti-Tubulin (Cat. T5168, Sigma, 1:2000).

What is the effect of vha-1 RNAi alone on translation in a polysome profiling experiment?

Answer: The primary objective of this experiment was to study the effect of *vha-1* RNAi on the translation of *atfs-1* during mitochondrial stress, we therefore did not include the *vha-1* RNAi data. In the revised manuscript, we have now included the *vha-1* RNAi control data (updated Fig. 4G and 4H). Based on the ribosomal profiling data (Updated Fig. 4G), *vha-1* RNAi alone generally reduced translation.

4. Effects of atfs-1 translation: ATFS-1 abundance is increased by its mRNA translation even though polysomes are mildly depleted. To better understand this, it would be important to know the translational efficiency of the investigated mRNAs (polysome-associated vs total mRNA). I find it difficult to understand why H1 translation would be affected by translation when it is controlled under the atfs-1 promoter. Is the translational effect specific to atfs-1 or to all mRNAs? How could the promoter affect the protein's biosynthesis? Is the H1 construct perhaps cloned in a way that it puts H1 under the control of ATFS-1 translation initiation sequence, or is it trans-spliced? If so, this would be an opportunity to study atfs-1 translation.

Answer: The *atfs-1p::Histone-wCherry* transgenic strain was constructed such that the expression of the Histone-mCherry reporter protein is under the strict control of the upstream intergenic sequences (including both the promoter and 5'-UTR regions) of *atfs-1* (PMID: 22508763). Therefore, similar to the ATF4 translation sensors constructed in mammalian systems (PMID: 30088945), the expression/translation of this Histone-mCherry is controlled in a similar fashion as that of the endogenous ATFS-1 protein. In addition, since the translated Histone-mCherry is not degraded by LONP-1 or other ATFS-1-specific-targeting enzymes, the *atfs-1p::Histone-wCherry* transgenic strain is therefore an ideal system to study ATFS-1 translation regulation, independent of its degradation. The data with the Histone-mCherry protein (controlled by the 5'-UTR of *atfs-1*) that exhibited a similar expression pattern as the ATFS-1::GFP reporter (Fig. 4D-4F), hence fully support our current model that the increased expression of ATFS-1 protein under mitochondrial stress is driven by the v-ATPase-TORC1-mediated translational regulatory mechanism, independent of the degradation of ATFS-1. We have now elaborated more on the specifics of the *atfs-1p::Histone-wCherry* transgenic strain in the revised manuscript to clarify this point (Page 9, Line 11).

5. To assess the processes contributing to ATFS-1 abundance, it would be important to consider its degradation as well.

Answer: We thank the reviewer for this comment. Our data that RNAi of *vha-1*, *vha-4*, *vha-16* or *vha-19* blocked the accumulation of ATFS-1 induced by mitochondrial stress even when *lonp-1* was silenced (updated Fig. 3J and S2I), suggested that v-ATPase regulated ATFS-1 protein expression independent of its degradation by LONP-1. In addition, as discussed in point #4, in the *atfs-1p::Histone-wCherry* transgenic strain, we found that the Histone-mCherry protein (controlled by the 5'-UTR of *atfs-1*) exhibited similar pattern of the ATFS-1::GFP reporter (Fig. 4D-4F). Collectively, these results fully supported our model that the increased expression of ATFS-1 protein under mitochondrial stress is mainly driven by the v-ATPase-TORC1-mediated translational regulation mechanism independent of its degradation, as further clarified in the revised manuscript (Page 9, Line 14).

Minor points:

As the authors suggest, it remains unclear how mito stress activates TOR signaling. I find it interesting that mito stress activates a "growth state" by triggering TOR. Activating TOR during stress seems to be a risky maneuver for cells. Moreover, I find it difficult to square the facts that TOR signaling is up during mito stress and that mito stress inhibits translation. It would be important to discuss this further, also in the light of the effect on survival: TOR

inhibition extends survival but so does mito stress, which activates TOR. It would be important to further highlight these points in the discussion and to include a section discussing the limitations of the study.

Answer: We thank the reviewer for these important comments/suggestions. As suggested, we have now further highlighted these points in the revised manuscript and included a section discussing the limitations of the study (Page 13, Line 23).

Reviewer #3:

Here, Li and colleagues searched for mediators of UPR^{mt} and found that RNAi of several vacuolar H⁺-ATPase components blocked expression of the UPR^{mt} reporter *hsp-6p::gfp* upon *cco-1* RNAi in a dose-dependent manner, and supporting evidence includes knockdown of other mitochondrial stress inducers, UPR^{mt} activators, and knockdown of v-ATPase function, but separate from other UPR pathways and separate from mitophagy/autophagy. The expression of ATFS-1 itself seems to be regulated by v-ATPase, since *cco-1*(RNAi) induces ATFS-1 protein but is blocked by knockdown of *vha* components. Linking to mTORC1, phosphorylation of RSKS-1 increases in a V-ATPase-dependent manner. V-ATPase regulation of ATFS-1 protein appears to be upstream of Lon protease function; therefore, the regulation of ATFS-1 protein must be at the translation level. Cytosolic ribosomal components are required for the ATFS-1 translation. The pathway also seems to be distinct from GCN-2, PEK-1, and EIF-2a kinase regulation. Overall, the experiments are presented well and are logical, and describe a new, TORC1-specific pathway of UPR^{mt} activation that, surprisingly, has not been previously described. The only major shortcomings are the lack of information on the nature of the signal from the damaged mitochondria, and information on whether translation is specifically and only of ATFS-1, or if there is a suite of proteins whose translation is activated in this manner.

Answer: We thank the reviewer for finding our work well-presented, logical and novel. We would also like to express our sincere appreciation for the constructive comments. We have now provided our answers to each of the comment as below and also adapted the content of our manuscript accordingly.

Major questions/suggestions:

1. Why do the different components not all have the same effect - is that an RNAi artifact (some clones don't work as well?), or some functional difference between the subunits? While I first thought the former must be true, the fact that the RNA-seq identifies expression differences that seem specific to the subunit knocked down is surprising. I realize that the 4-way Venn is appealing, but perhaps the proportional Venn diagram (using boxes) might better convey the fact that the majority of the *cco-1*-induced v-ATPase-dependent genes (325) are in fact shared - and thus suggests that the differences might be minor and due to RNAi efficacy differences. (The PCA also emphasizes differences, but might be unavoidable.)

Answer: We thank the reviewer's insightful comments. We agree that some v-ATPase subunit RNAi failed to attenuate UPR^{mt} is probably due to their relatively weak knockdown efficiency, especially considering that in our screen system, only 60% of these v-ATPase RNAi bacteria were used (since *cco-1* RNAi occupied the rest 40%), and some RNAi clones may thus not work so efficiently in disrupting overall v-ATPase activity. Indeed, as also pointed by Reviewer #1 (the 1st comment), RNAi of *unc-32*, *vha-7*, *vha-14*, and *vha-18* has been shown to suppress the UPR^{mt} induction in *drp-1* mutants (PMID: 32737159). To clarify this discrepancy, we reduced the *cco-1* RNAi occupancy to 20%; allowing to increase the RNAi of *unc-32*, *vha-7*, *vha-14*, and *vha-18* to 80%. As shown in the new Figure S1A, we found that 80% RNAi of *unc-32*, *vha-7*, *vha-14*, and *vha-18* indeed attenuated the UPR^{mt} activation by *cco-1* (20%) RNAi to some extent. Furthermore, the effect of *vha-14* and *unc-32* RNAi on UPR^{mt} suppression was more potent than that of *vha-7* and *vha-18* RNAi, in line with the results in the *drp-1* mutants (PMID: 32737159, Suppl. Fig. 4C). These results have now also been included in the revised manuscript (Page 4, Line 29). As for the Venn diagram in Fig. 2C, to better convey the message that the majority of the *cco-1* RNAi-induced v-ATPase-dependent genes (325) are in fact shared, we have used an UpSet plot to present this data (New Fig. 2C).

2. Figure 6 - I understand that *isp-1* and *clk-1* are mutants and therefore could be treated with *vha* RNAis to check growth, but perhaps images of the double RNAi of *cco-1* and *vha* could be shown here as well in Fig. 6A (i.e., like Fig. 1A but in N2 rather than *hsp-6p::gfp*) - easy to add these photos since the lifespans are already shown.

Answer: We thank the reviewer for this great suggestion. As expected, we acquired these data, which have now been included in the revised manuscript as Fig. S5B.

3. The model as shown makes sense, but what is the signal from stressed mitochondria to RHEB and TORC1 to result in such a specific translation of ATFS-1? The authors acknowledge this gap, but do the authors think there is a specific peptide that is transported to the lysosomes that somehow activates RHEB, or is it a general response to increased vacuolar occupancy upon increased unfolded proteins upon mitochondrial stress? Could they test the contact site model?

Answer: To investigate the exact mechanism on how the signal from stressed mitochondria to RHEB and TORC1 results in such a specific translation of ATFS-1, and how the contact site model fit in our story is likely beyond the scope of the current study, which aimed to reveal the key role of v-ATPase/mTORC1-mediated ATFS-1 translation in UPR^{mt} activation. Nevertheless, we have used another lysosomal acidification inhibitor chloroquine (CQ), and found that CQ inhibited the UPR^{mt} in a dose-dependent manner (new Fig. 1H and S2D), suppressed TORC1 activity as well as the accumulation of ATFS-1 upon mitochondrial stress (new Fig. S2E). These results suggest that the vacuolar pH and intact lysosomal function are required for TORC1 and UPR^{mt} activation. The new data with CQ have now also been included and discussed in the revised manuscript (Fig. 1H, S2D and S2E; Page 5, Line 11; Page 7, Line 31). In addition, we have included a section discussing the limitations of the current study and the unsolved questions that need to be addressed in future work (Page 13, Line 23).

4. Similarly, the specific activation in translation of ATFS-1 seems solid. Are there other proteins that are also increased by translation activation along with ATFS-1 as a result of this TORC-1-mediated translation activation?

Answer: We thank the reviewer for this comment, we have now further measured the polysomal level of more genes/transcription factors involved in different stress responses, including *dve-1*, *rsk-1*, *xbp-1s*, *hsp-16.2*, *hsp-3*, *sod-3* (New Fig. S3, D and F). These results indicate that only *atfs-1* and its downstream UPR^{mt} genes (e.g., *hsp-6* and *gpd-2*) are upregulated in translation upon mitochondrial stress and in a VHA-1/TORC1-dependent fashion.

5. Is there a way to quantitate and compare the magnitudes of the effects of TORC1-mediated ATFS-1 translation vs ATFS-1 nuclear translocation into the nucleus/Lon protease mechanism, so we can understand their relative importance?

Answer: To quantitate and compare the magnitudes of the effects of the two mechanisms in UPR^{mt} activation is very challenging, especially considering that single inhibition of the Lon protease is in fact insufficient to activate the UPR^{mt} (Fig. 3, C and D), and increased translation of ATFS-1 might also become non-functional if the mitochondrial stress-induced nuclear accumulation of ATFS-1 is not inhibited during mitochondrial stress. Notably, by using the *atfs-1p::atfs-1::flag::gfp* strain (OP675: <https://cgc.umn.edu/strain/OP675>) with a EGFP::3xFLAG tag inserted in frame at C-terminus of the genomic locus of *atfs-1*, we found that mitochondrial stress-induced nuclear accumulation of ATFS-1::GFP was not detected in worms with the co-treatment of *vha-1*, *vha-4*, *vha-16* and *vha-19* RNAi (new Fig. 2I). This result suggested that v-ATPase/TORC1-mediated translation of ATFS-1 is a prerequisite step for the increased accumulation and the subsequent nuclear-localization of ATFS-1 for UPR^{mt} activation in response to mitochondrial stress, suggesting that the two mechanisms likely act as a whole to ensure that enough ATFS-1 is translocated into the nucleus for UPR^{mt} activation. The related discussion has also been added in the revised manuscript (Page 12, Line 13).

Minor points (writing suggestions):

1. "Despite that some evidence implicated the involvement of TORC1 components" - revise to "Despite some evidence that implicated" or "Despite the fact that some evidence implicated"
2. "majorly" would read better as "primarily"
3. "mechanism on how TORC1"  "mechanism of how" (this appears several times in the paper)
4. "little is known on how"  "little is known about how" or "little is known of how"
5. "Furthermore, whether the communications between" \diamond communication (does not need to be written as plural)

Answer: We thank the reviewer for these suggestions (#1-5), and we have adapted them in our revised manuscript accordingly.

6. It might be better to group all of the initial TOR pathway results together (*RSKS-1*, *let-363*, *rheb-1*, *Torin1*), rather than breaking them up with the *lonp-1* results in between, if there is no reason to split them (I can't see an obvious reason why Lon is put before the other TOR components - if there is a reason they are separated, make it clearer to the reader).

Answer: We thank the reviewer for this comment, and have now grouped all of the initial TOR pathway results together as suggested.

October 4, 2022

RE: JCB Manuscript #202205045R

Dr. Johan Auwerx
École Polytechnique Fédérale de Lausanne
Lab of Integrative Systems Physiology SV/IBI1/LISP/NCEM Station 15
Lausanne CH1015
Switzerland

Dear Johan,

Thank you for submitting your revised manuscript entitled "V-ATPase/TORC1-mediated ATFS-1 translation directs mitochondrial UPR activation in *C. elegans*". We would be happy to publish your paper in JCB pending final revisions necessary to meet our formatting guidelines (see details below).

A. MANUSCRIPT ORGANIZATION AND FORMATTING:

- 1) Text limits: Character count for Articles is < 40,000, not including spaces. Count includes abstract, introduction, results, discussion, and acknowledgments. Count does not include title page, figure legends, materials and methods, references, tables, or supplemental legends.
- 2) Figures limits: Articles may have up to 10 main text figures.
- 3) Figure formatting: Scale bars must be present on all microscopy images, including inset magnifications. Molecular weight or nucleic acid size markers must be included on all gel electrophoresis.
- 4) Statistical analysis: Error bars on graphic representations of numerical data must be clearly described in the figure legend. The number of independent data points (n) represented in a graph must be indicated in the legend. Statistical methods should be explained in full in the materials and methods. For figures presenting pooled data the statistical measure should be defined in the figure legends. Please also be sure to indicate the statistical tests used in each of your experiments (either in the figure legend itself or in a separate methods section) as well as the parameters of the test (for example, if you ran a t-test, please indicate if it was one- or two-sided, etc.). Also, if you used parametric tests, please indicate if the data distribution was tested for normality (and if so, how). If not, you must state something to the effect that "Data distribution was assumed to be normal but this was not formally tested."
- 5) Abstract and title: The abstract should be no longer than 160 words and should communicate the significance of the paper for a general audience. The title should be less than 100 characters including spaces. Make the title concise but accessible to a general readership.

* Please note that our typical guideline is to not include the species name in a title unless necessary
- 6) Materials and methods: Should be comprehensive and not simply reference a previous publication for details on how an experiment was performed. Please provide full descriptions in the text for readers who may not have access to referenced manuscripts.
- 7) Please be sure to provide the sequences for all of your primers/oligos and RNAi constructs in the materials and methods. You must also indicate in the methods the source, species, and catalog numbers (where appropriate) for all of your antibodies. Please also indicate the acquisition and quantification methods for immunoblotting/western blots.
- 8) Microscope image acquisition: The following information must be provided about the acquisition and processing of images:
 - a. Make and model of microscope
 - b. Type, magnification, and numerical aperture of the objective lenses
 - c. Temperature
 - d. Imaging medium
 - e. Fluorochromes

f. Camera make and model

g. Acquisition software

h. Any software used for image processing subsequent to data acquisition. Please include details and types of operations involved (e.g., type of deconvolution, 3D reconstitutions, surface or volume rendering, gamma adjustments, etc.).

10) Supplemental materials: There are strict limits on the allowable amount of supplemental data. Articles may have up to 5 supplemental figures. Please also note that tables, like figures, should be provided as individual, editable files. A summary of all supplemental material should appear at the end of the Materials and methods section.

13) ORCID IDs: ORCID IDs are unique identifiers allowing researchers to create a record of their various scholarly contributions in a single place. At resubmission of your final files, please consider providing an ORCID ID for as many contributing authors as possible.

Please note that JCB now requires authors to submit Source Data used to generate figures containing gels and Western blots with all revised manuscripts. This Source Data consists of fully uncropped and unprocessed images for each gel/blot displayed in the main and supplemental figures. Since your paper includes cropped gel and/or blot images, please be sure to provide one Source Data file for each figure that contains gels and/or blots along with your revised manuscript files. File names for Source Data figures should be alphanumeric without any spaces or special characters (i.e., SourceDataF#, where F# refers to the associated main figure number or SourceDataFS# for those associated with Supplementary figures). The lanes of the gels/blots should be labeled as they are in the associated figure, the place where cropping was applied should be marked (with a box), and molecular weight/size standards should be labeled wherever possible.

B. FINAL FILES:

**The license to publish form must be signed before your manuscript can be sent to production. A link to the electronic license to publish form will be sent to the corresponding author only. Please take a moment to check your funder requirements before

choosing the appropriate license.**

Thank you for this interesting contribution, we look forward to publishing your paper in Journal of Cell Biology.

Sincerely,

Jodi

Jodi Nunnari, Ph.D.
Editor-in-Chief

Andrea

Andrea L. Marat, Ph.D.
Senior Scientific Editor

Journal of Cell Biology

Reviewer #2 (Comments to the Authors (Required)):

The authors have addressed the points I raised and I gladly support publication of the manuscript.